# Probing the zooarchaeological record across time and space for ancient pathogen DNA

Anne Kathrine W. Runge [1] ✉, Ian Light-Maka [1,2], Ken Massy [3], Marcel Keller [4,5], Simon Trixl [6,7], Helja Kabral [4], Casey L. Kirkpatrick [8,9,10], Kirsten Bos [9], Jana Eger [11], Michal Ernée [12], René Kyselý [13], Michael Hochmuth[14], Dominik Poradowski[15], Aleksander Chrószcz[15], Norbert Benecke[16], David Daněček [17], Jana Klementová[17], Anatoli Nagler[16], Alexey A. Kalmykov [18], Anatoly R. Kantorovich[19], Vladimir E. Maslov[20], Andrey B. Belinskiy[21], Christiana L. Scheib [4,22], Meda Toderaş[23], Svend Hansen [16], Philipp W. Stockhammer [3,9], Kai Kaniuth [24], Regina Uhl[16], Sabine Reinhold [16], Rosalind E. Gillis [14], Elizabeth A. Nelson[25], Kamilla Pawłowska [26] ✉ & Felix M. Key [1] ✉

Zoonoses are among the greatest threats to human health, with many zoonotic pathogens believed to have emerged following the Neolithic transition. Palaeomicrobiological investigations of the zooarchaeological record hold potential to uncover the reservoirs, host ranges, and host adaptations of zoonotic pathogens in the past, but face challenges in identifying promising specimens and pathogen DNA preservation. We perform palaeopathological and genetic examinations of 346 skeletal elements from domesticated and wild animals collected from 34 Eurasian sites dating across the last six millennia. We identify 116 signatures of 29 ancient (opportunistic) pathogens and find support that palaeopathological lesions provide guidance for specimen selection. For two pathogen species, *Erysipelothrix rhusiopathiae* and *Streptococcus lutetiensis*, we confirm their ancient authenticity using phylogenetics, showcasing an approach to explore the relationship between ancient low-coverage genomes and their modern-day relatives. Our work presents a pathway to understanding prehistoric zoonotic diseases by integrating zooarchaeological, palaeopathological, and genetic data.

Infectious diseases are a major health concern responsible for an estimated 13.7 million deaths worldwide in 2019[1]. Approximately 60% of human pathogens are believed to have originated from animals with dramatic consequences throughout history[2]. For instance, the 1918 flu pandemic likely had an avian origin[3], while outbreaks of plague, which are transmitted from rodent hosts via flea vectors[4], led to multiple pandemics, including the Black Death in 1346–1353 CE[5]. Although Palaeolithic hunter-gatherers came into contact with diseased animals[6], many zoonotic diseases were likely introduced into human populations following the introduction of farming during the Neolithic

Transition starting around 12,000 years ago[7,8]. The Neolithic period introduced complex changes in the spatial organisation of settlements, demographic expansion, and subsistence patterns, which favoured the introduction and persistence of novel infectious diseases[9,10]. In addition, the intensification of inter-species interactions between humans and their domesticated animals during livestock management, butchering, or consumption of animal-derived products increased the likelihood of zoonotic, reverse zoonotic, and animal-to-animal transmission events[7,11]. Today, this is recognised in the One Health concept arguing that human health is interconnected with

animal and environmental health with consequences ranging from zoonoses to antimicrobial resistance to food safety[12]. Ancient DNA holds promise to be a powerful tool for exploring the One Health concept in deep time - recently also coined One Palaeopathology[13] - by reconstructing pathogen genomes from ancient human and animal remains, as well as the environment, revealing prehistoric disease reservoirs, retracing pathogen distribution, and uncovering adaptations to the human host.

While screening for microbes in large genomic datasets generated for ancient human population studies has become reasonably routine, the same cannot be said for ancient faunal datasets. As a result, limited information is available concerning disease reservoirs in ancient and historic animal populations - posing a barrier to integrate the One Health concept and ancient DNA. Nevertheless, recent advances provided some notable exceptions. For example, a *Yersinia pestis* genome from Bronze Age domesticated sheep provided insights into the host range and evolution of a pathogen lineage thus far exclusively identified in human archaeological remains[14]. Furthermore, an 8000 year old *Brucella melitensis* genome from a sheep confirms the emergence of the zoonosis Brucellosis during the Neolithic period[15], a Medieval strain of *Mycobacterium leprae* revealed reverse zoonosis in squirrels[16], and reconstructed Marek disease virus genomes informed upon the origin and virulence of a contemporary livestock infection[17]. These studies highlight the power of pushing palaeomicrobiology into the zooarchaeological record to explore disease reservoirs, host ranges, and evolution in ancient and historic animal populations. However, so far no systematic investigation has been conducted to test the recovery of microbial pathogen genomes directly from zooarchaeological remains.

Identifying promising specimens for DNA sampling presents a major challenge in recovering microbial pathogen DNA from faunal remains. One of the primary complications is that the majority of animals under human management died from slaughter rather than natural causes, including infectious diseases, while potential culling of sick animals would further reduce the likelihood of diseased specimens being preserved. In addition, animal remains may have experienced amplified DNA degradation due to heating, boiling, or roasting during preparation for consumption[18]. Moreover, unlike human remains, which were more frequently intentionally buried, the majority of animal remains were discarded as household waste, resulting in prolonged exposure to the environment and thus degrading target DNA[18]. Altogether these factors likely decrease the probability of recovering ancient pathogen DNA from animals, explaining why only a few studies have described pathogen genomes reconstructed from faunal remains.

Sometimes evidence of disease is preserved within the archaeological record in the form of palaeopathological lesions[19,20]. Such lesions typically occur as a result of prolonged disease, but while they are frequently recorded and studied in human remains, the same is rarely the case for animals[19]. Animal palaeopathology differs from human palaeopathology, both in terms of material studied, but also in the conclusions that can be drawn from morphological analysis alone. A key challenge is the disarticulated and fragmented nature of most zooarchaeological deposits, which precludes the systematic, skeleton-wide palaeopathological analyses typically performed on human remains[20,21]. Moreover, skeletal pathologies in animals have historically been understudied, with limited support from modern data, leading to continued reliance on assumptions derived from human pathological conditions[20,21]. As a result, palaeopathological analyses of animal remains are largely limited to isolated skeletal elements, which restricts analyses to the characterisation of a general health status. Together, these factors complicate the formulation of a differential diagnosis, thus preventing the identification and study of specific diseases within and across populations. Compounding this issue, few zoonotic infections leave diagnostic traces in animal bone

assemblages while bone modifications arising from arduous animal treatment can appear similar to genuine palaeopathological changes, making infectious disease in animal populations difficult to identify through morphological analysis alone[22]. Nevertheless, once identified, they may allow researchers to overcome, at least in part, the challenges of palaeomicrobiology in the zooarchaeological record, enabling DNA analyses to identify the pathogens present in prehistoric animal populations.

In this study, we investigate whether a selective sampling approach based on palaeopathological analysis and teeth can provide useful targets for ancient pathogen DNA sampling and overcome the challenges in faunal palaeomicrobiology. We investigate 346 zooarchaeological specimens from across Eurasia for pathogen DNA, with a particular focus on the Bronze Age, a promising period for ancient zoonotic pathogen recovery based on human pathogen investigations. Our results show ancient DNA signatures of known (opportunistic) pathogens in domesticated animals, and highlight the power of palaeopathology in prioritising specimens for pathogen DNA recovery, opening a new direction in palaeomicrobiology.

## Results

### Zooarchaeological collection of 346 skeletal elements from across Eurasia

To understand whether palaeopathological lesions or teeth provide useful targets for recovering ancient microbial pathogen DNA from zooarchaeological remains, we collected a total of 346 skeletal elements from at least 328 individual animals discovered at 34 archaeological sites across Eurasia (Fig. 1a, Supplementary Data 1). The majority of these sites are located in Europe, but two sites, Monjukli Depe and Tilla Bulak, are located in Central Asia. The sites span approximately 5800 years of human history (Fig. 1b), with the oldest site, Monjukli Depe, dated to 4650–4350 BCE and the youngest site, Giecz 10, dated to 900–1200 CE. The sites cover periods dating from the Neolithic to the Medieval period, with the majority of sites belonging to the Bronze Age (Supplementary Data 1). These periods took place at different absolute chronological horizons due to spatiotemporal differences in material culture development within discrete geographical locations. We focused on Bronze Age specimens because human-derived ancient zoonotic pathogen genomes have been repeatedly identified in the literature in specimens from this period, a time of major human migratory events following the Neolithisation in Eurasia[23–28].

The specimen selection was aimed primarily at domesticated species, but wild animals were also included, especially if their abundance indicated that they contributed significantly to the local economy. All specimens were taxonomically assigned based on morphology and later also classified using the ancient host DNA content of the generated metagenomic data (see below for data generation). Without considering articulation, we collected a total of 112 cattle, 94 sheep, 48 pigs/wild pigs, 18 dogs, and 11 goats along with 48 remains from species that were less abundant than those previously mentioned or could not be classified to a lower taxonomic unit (Fig. 2, Supplementary Data 2).

### Palaeopathological and taphonomic evidence in the zooarchaeological record

In total, we collected 188 bones with palaeopathological lesions (54.6%), 27 bones with no identified palaeopathological changes (7.2%), and 131 teeth (38.2%) from the 34 sites included in this study. From one individual from Romania (Pietrele) and seven of the German individuals, we collected both bones and one or more teeth. Bones from the sites in Germany, Poland, Romania (Pietrele), and Uzbekistan (Tilla Bulak) were visually inspected for palaeopathological lesions (Fig. 3, Supplementary Data 2). Although teeth can have lesions such as caries, none were observed in the studied material, and none of the

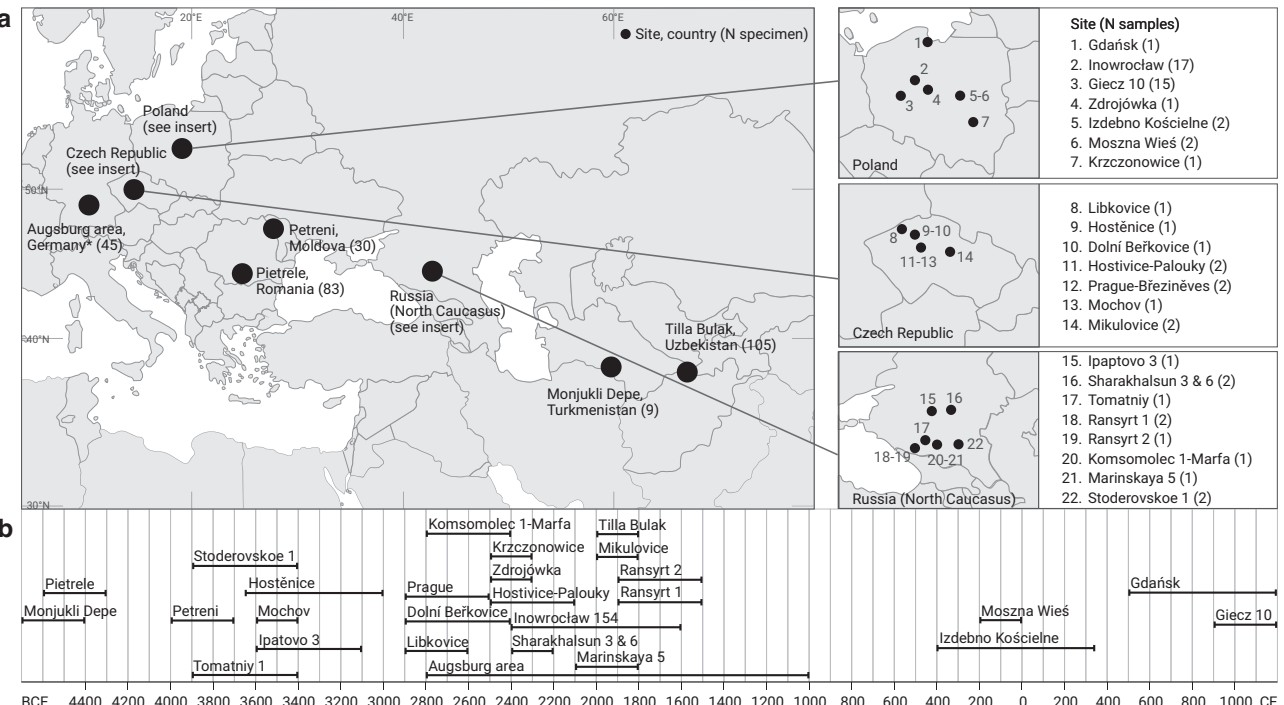

**Fig. 1 | Geographical and temporal distribution of 346 zooarchaeological specimens. a** The 34 archaeological sites included in this study are distributed across Europe and Central Asia. Cut-out maps are included for countries with more than one site. *The Augsburg area (Germany) consists of seven sites in and around Augsburg, which are not shown individually on the map. Coordinates are available in Supplementary Data 1. **b** The timeline shows the age ranges of each site included in this study, which are estimates based on C14 dates of associated specimen or archaeological dating based on features (Supplementary Data 1 and Supplementary Information). Note that the Bronze Age covers different absolute chronological horizons within different geographical locations.

teeth included in the study displayed pathological conditions. The investigated specimens were deposited in sand or silt, and some had gypsum encrustations, which did not prevent conducting the palaeopathological analysis. Identification of skeletal signs of infectious disease among the zooarchaeological specimens can be difficult to discern from unrelated taphonomic processes[29] or conditions arising from animal labour. We macroscopically screened the remains for various conditions that can be associated with, though not necessarily unique to, infection. This included: periostitis (inflammation of the periosteum, a connective tissue layer on the surface of bones, that is sometimes caused by infection; Fig. 3a, d), macroporosity suggestive of inflammation (Fig. 3b, c), skeletal trauma (which may create entry sites for pathogens), lytic lesions and/or pathological new bone formation (sometimes associated with chronic infectious disease), alveolar recession (indicative of periodontitis, a severe gum infection), ante-mortem tooth loss (also sometimes associated with oral infections, such as periodontitis, caries and/or abscess; Fig. 3e), and arthropathy (sometimes related to infectious disease)[19].

Across all initially investigated sites only a minority of the zooarchaeological deposits showed any palaeopathological evidence but those included a variety of lesion types affecting a wide range of skeletal elements in many species (Fig. 3, Supplementary Data 2). More specifically, the bone assemblages from Poland, Romania (Pietrele), and Uzbekistan (Tilla Bulak) showed pathological lesions including periostitis, osteolytic lesions, oral pathologies, and arthropathy. Only some of the specimens from Germany (Augsburg area) displayed palaeopathological changes, including evidence of periostitis, osteomyelitis, and periodontitis, while the majority of collected specimens had no visible lesions. Across the investigated sites, a wide range of primarily domestic but also wild species is represented among the specimens with palaeopathological lesions including: horses, cattle, pigs, sheep, goats, dogs, red deer, aurochs, gazelles, and a fox and a

beaver. Overall, sheep were dominant among the species showing palaeopathological lesions in our dataset (68 of the total 187 pathological specimens), but not significantly overrepresented after multiple testing corrections (Bonferroni method). Similarly, lesions were identified on a wide range of skeletal elements with mandibles ($n = 25$), ribs ($n = 24$), and vertebrae ($n = 19$) being the majority. Collectively, the zooarchaeological specimen collection included a vast range of species and bone types covering different palaeopathological lesions suggestive of infection and thereby pathogens whose DNA fingerprint we might recover.

## Identification and authentication of ancient pathogen signatures

From the 346 skeletal elements, we produced a total of 357 DNA extracts. These included 20 subsamples collected from nine bones as well as 11 teeth collected from four individuals (see Supplementary Data 2 for further details). Samples were sequenced on the Illumina platform, which, apart from five failed samples, generated between 2,324,302 and 90,823,912 sequencing reads per sample (Supplementary Data 2). All sequencing reads which passed quality control thresholds and did not map to the human reference genome (see Methods) were taxonomically classified. The presence of DNA molecules from a range of bacterial, viral, and parasitic species was investigated (Supplementary Data 3) using the HOPS pipeline[30] as implemented in nf-core/eager[31]. We focus our analysis on species that are pathogenic to humans and animals, including species that are opportunistic pathogens able to colonise without causing disease in the host. The HOPS pipeline enables the identification of target DNA in a metagenomic dataset and at the same time interrogates signatures for its ancient authenticity, for example mismatches due to deamination at terminal bases of DNA fragments (ancient DNA damage)[32] or distribution of aligned reads along the genome[33]. We applied stringent

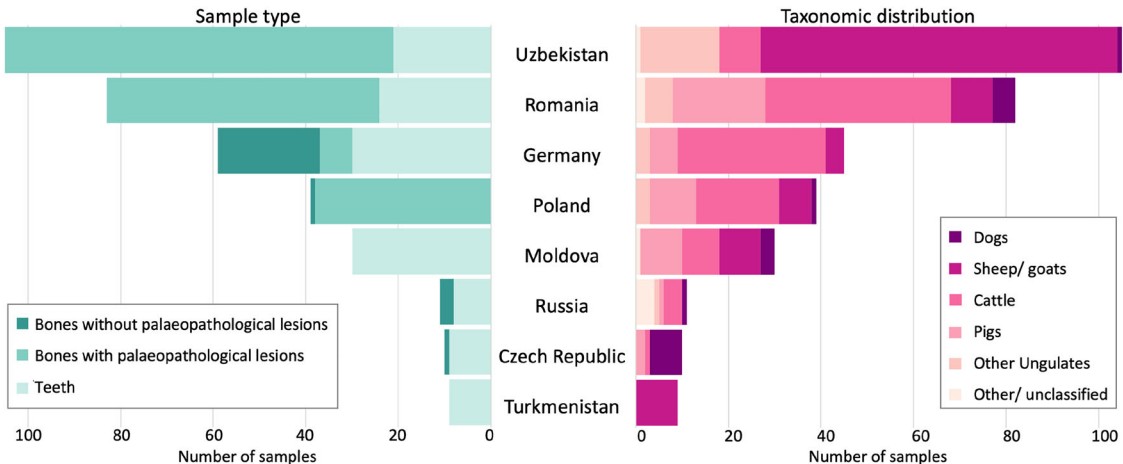

**Fig. 2 | Sample type and taxonomic distribution across countries.** Left: Showing the distribution of the 346 skeletal elements (bones with palaeopathological lesions, bones with no lesions, and teeth) that were sampled at each country. Right: Taxonomic distribution of the 331 sampled individual animals across countries illustrating that the majority of samples originated from domesticated species. For the taxonomic distribution, mandibles where both bone and teeth were sampled (Pietrele and Germany) are counted as one sample, while potentially articulating bones are considered separately. The 'other ungulates' category includes ungulates other than sheep, goat, and cattle as well as ungulates that could not be classified at the species level. The 'other/ unclassified' category includes all other taxonomic groupings than those already mentioned. See Supplementary Data 2 and Source Data 1 for detailed taxonomic classification of all samples.

authentication criteria to ensure that any pathogens identified in the screening dataset were robust. These included a minimum of 50 reads aligning to the target taxonomic node, terminal bases showing a minimum of 10% DNA damage (for single-stranded libraries, 5' or 3' terminal C to T; for double-stranded libraries C to T at the 5' terminal base or G to A at the 3' terminal base), a declining edit distance, and little to no read stacking except for samples with exceptionally high number of reads (Methods).

From our screening dataset, 55 libraries (15%) produced high confidence bacterial hits that met our identification and authentication criteria (Fig. 4). 30 samples produced hits to only one bacterial species, while 25 samples produced hits to more than one species. No hits to eukaryotic parasites or viruses were identified in our dataset.

The authenticated bacterial species differed in their described pathogenic potential with some being identified as primarily pathogenic while others are opportunistic pathogens that can also asymptomatically colonise the host. The mostly pathogenic bacteria included the species *Salmonella enterica, Bordetella petrii, Coxiella burnetii*, and *Erysipelothrix rhusiopathiae* (Fig. 4). *S. enterica* is a major zoonotic pathogen consisting of hundreds of serovars, with different host specificities of which some are known to infect humans causing gastroenteritis or systemic disease[34,35]. Bacterial hits from four samples passed our stringent authentication criteria: AZP-056, a sheep tooth from Petreni (Romania), AZP-116, a sheep mandible with periodontal disease from Pietrele, AZP-148, a dog tooth from Pietrele, and AZP-277, a sheep femur with arthropathies from Tilla Bulak. *Bordetella* species are frequent pathogens of humans and animals, but the clinical relevance of *B. petrii* (recovered from AZP-156, a dog mandible with oral pathology from Pietrele) is unclear[36]. *C. burnetii* is an obligate intracellular pathogen, which causes the disease coxiellosis in animals although infected individuals are often asymptomatic[37]. We obtained a hit to *C. burnetii* from sample AZP-247, a dog milk tooth from Tilla Bulak. *E. rhusiopathiae* infections have been characterised in a wide range of vertebrate and invertebrate species[38], and was recently identified in human remains from medieval Southwest Europe[39]. It is the etiologic agent of swine erysipelas[40], but we identify it in cattle (AZP-012, a tooth from Marinskaya 5, and AZP-183, an astragalus with a porous bone structure potentially associated with inflammatory disease from Pietrele) where it is known as an opportunistic pathogen that has been associated with septicaemia, abscesses in the liver and

lungs, encephalomeningitis, polyserositis and septic arthritis[41]. The other identified bacterial species belong to the oral and gastro-intestinal microbiome. These include *Corynebacterium stationis, Corynebacterium xerosis, Escherichia coli, Porphyromonas gingivalis, Veillonella parvula, Yersinia intermedia*, as well as several species of *Enterococcus, Staphylococcus* and *Streptococcus*. While the majority of these species are members of the stable microbiome of animals or, when information on animals is limited, of humans, most are known opportunistic pathogens causing disease primarily among immuno-compromised individuals.

## Interpretation of DNA evidence in its palaeopathological context

Of the 55 samples that produced robust bacterial hits, 49 came from sites that were palaeopathologically investigated. Eleven of the 55 samples were teeth (8.4% of all teeth sampled), while the remaining 44 were bones with palaeopathological lesions (23.3% of all bones with lesions). Considering only the archaeological sites studied for palaeopathologies, this suggests that the selective sampling approach was more likely to yield DNA from pathogens ($p < 0.00105$, Fisher Exact test), a signal robust also when excluding teeth, which showed no pathologies ($p < 0.002$, Fisher Exact test). Although this study is biased towards bones with pathologies, the fact that no ancient pathogenic bacteria were authenticated from the 27 bone samples without any macroscopically detectable lesions ($p < 0.00035$, Fisher Exact test) emphasises the advantage of palaeopathological investigations for prioritisation of specimens. The majority of hits were to skeletal elements from sheep ($n = 31$). Other host species included cattle, gazelle, goat, wild or domesticated pig, dog, and the more generalised 'small ungulate' category. The lesions which produced bacterial hits were located on multiple types of skeletal elements, but most frequently on mandibles ($n = 8$), vertebrae ($n = 8$), and ribs ($n = 7$), which is consistent with the skeletal elements we most frequently collected during the palaeopathological selection.

The oral cavity is a reservoir of commensal bacteria that can turn into opportunistic pathogens, but it is also an entry gate for foreign bacteria[42]. The pathological changes in the mandibles that produced bacterial hits included periodontitis, abscesses, and antemortem tooth loss, with genomic signatures for *Bordetella, Corynebacterium, Enterococcus, Salmonella, Staphylococcus*, and *Streptococcus* (Fig. 4,

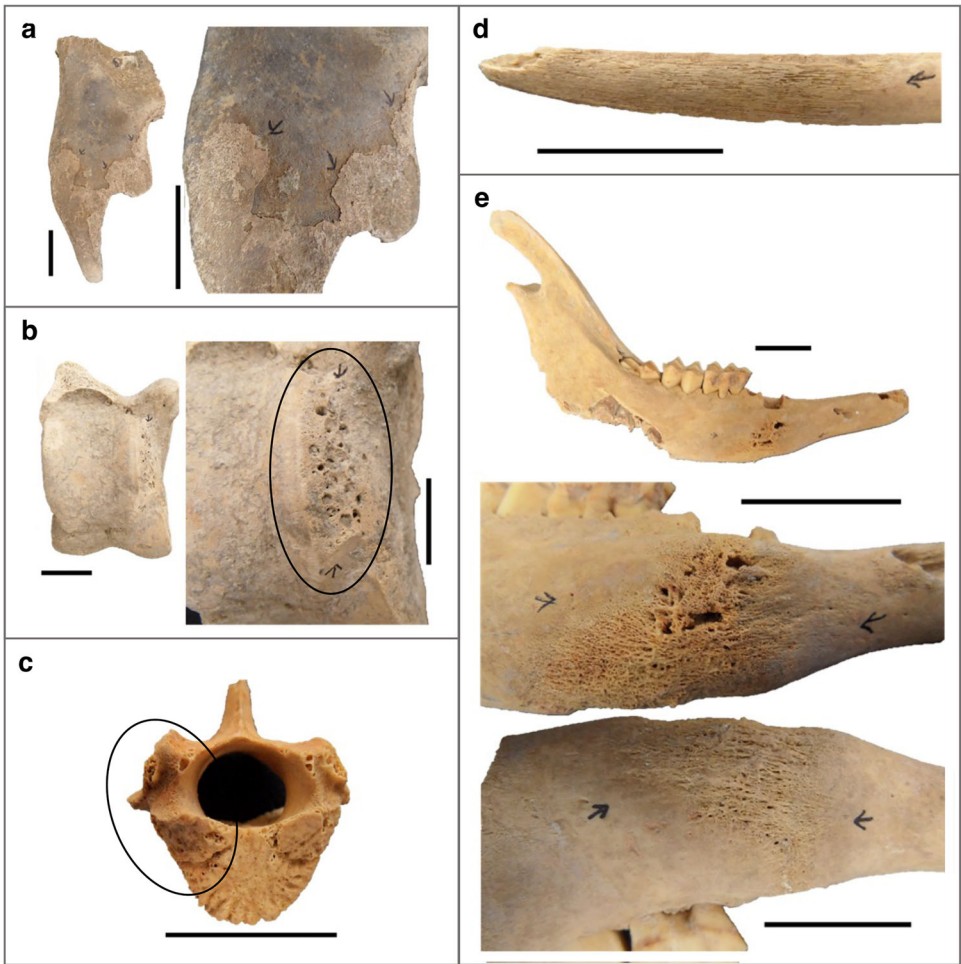

**Fig. 3 | Examples of palaeopathological lesions that were sampled for this study and produced pathogen DNA signatures in the screening.** Arrows or circles indicate location of lesions indicative for potential infection. **a** AZP-115, a pig skull fragment with woven periostitis, that produced hits to *Corynebacterium stationis* and *Streptococcus salivarius*. **b** AZP-183, a cattle astragalus with macroporosity, that yielded *Erysipelothrix rhusiopathiae* DNA. **c** AZP-268, a thoracic vertebra from a sheep with macroporosity on the vertebral body and part of the neural arch (circled), that produced hits to *Staphylococcus aureus*, *Staphylococcus cohnii*, *Staphylococcus equorum*, *Staphylococcus nepalensis*, and *Staphylococcus succinus*. **d** AZP-195, a small ruminant rib with periostitis, that yielded DNA from *Staphylococcus nepalensis*. **e** AZP-222, the sheep mandible shows marked, active infection, with reactive bone formation, an abscess, and early fistula formation, that produced hits to *Corynebacterium stationis*, *Staphylococcus nepalensis*, *Staphylococcus succinus* and *Streptococcus lutetiensis*. See Supplementary Data 2 for additional information. Scale bars indicate 2 cm.

Supplementary Data 2). Periodontitis, for example, is a polymicrobial disease that is often caused by members of the oral microbiome, known as the 'red complex', including *P. gingivalis*, which is the principal cause of this condition in humans[43] and known from broken-mouth periodontitis in sheep[44]. We recovered DNA from *P. gingivalis* from a sheep tooth, AZP-123, from Pietrele and a sheep radius, AZP-289, with periostitis from Tilla Bulak. While blood-borne pathogens are frequently recovered from human (brachydont) teeth[33], it is unclear if the same is true for species that do not have enclosed pulp chambers, such as ruminants (hypsodonts[45]). Of the 11 teeth that produced bacterial hits, four belonged to dogs (brachydont; 12.5% of total dog teeth), while 7 belonged to hypsodonts (cattle, sheep and goats; 8% of total teeth from those species), showing no significant difference in the identification of pathogen signatures among our dataset ($p = 0.09$, Fisher Exact test). In line with those results, none of the 14 pig molars (brachydont, although pig canines are hypsodont) produced robust hits.

Sampled postcranial skeletal elements were primarily ribs and vertebrae. Lesions on ribs could be indicative of pulmonary infections, and several of the bacterial species signatures ($n = 7$) recovered from lesions on ribs were species of *Staphylococcus* and *Streptococcus*, including *Streptococcus pneumoniae*, which colonise the airways and can cause infections in the lungs[46]. Other species signatures recovered from palaeopathological lesions on ribs included species of *Enterococcus* and *E. coli*, both potentially contaminants due to their abundance in human microbiomes. The eight vertebrae that produced bacterial hits displayed pathologies characteristic of inflammatory disease and infection, with genomic matches to *Corynebacterium*, *Staphylococcus* and *Streptococcus*.

In terms of geographic patterns, more than half of the specimens with identified bacterial hits originated from the Tilla Bulak site (2000–1800 BCE) in Uzbekistan (Central Asia). Although the specimens collected from this site constitute 29% of the entire dataset, the number of positive bacterial hits (58.2% of total samples with bacterial hits) is disproportionate compared to other sites ($p < 0.00002$; Fisher Exact test). No other sites had disproportionate pathogen recovery that rose to statistical significance. While we observe no correlation between the percentages of host and pathogen DNA recovered (Fig. S1) and host DNA preservation was overall low (median 0.38%; mean 1.16% of quality control (QC) passed reads), we identify, again, in samples from Tilla Bulak (27/105) as well as the Polish site Inowrocław (9/17) an elevated host DNA preservation of above 1% ($p < 0.00001$ and

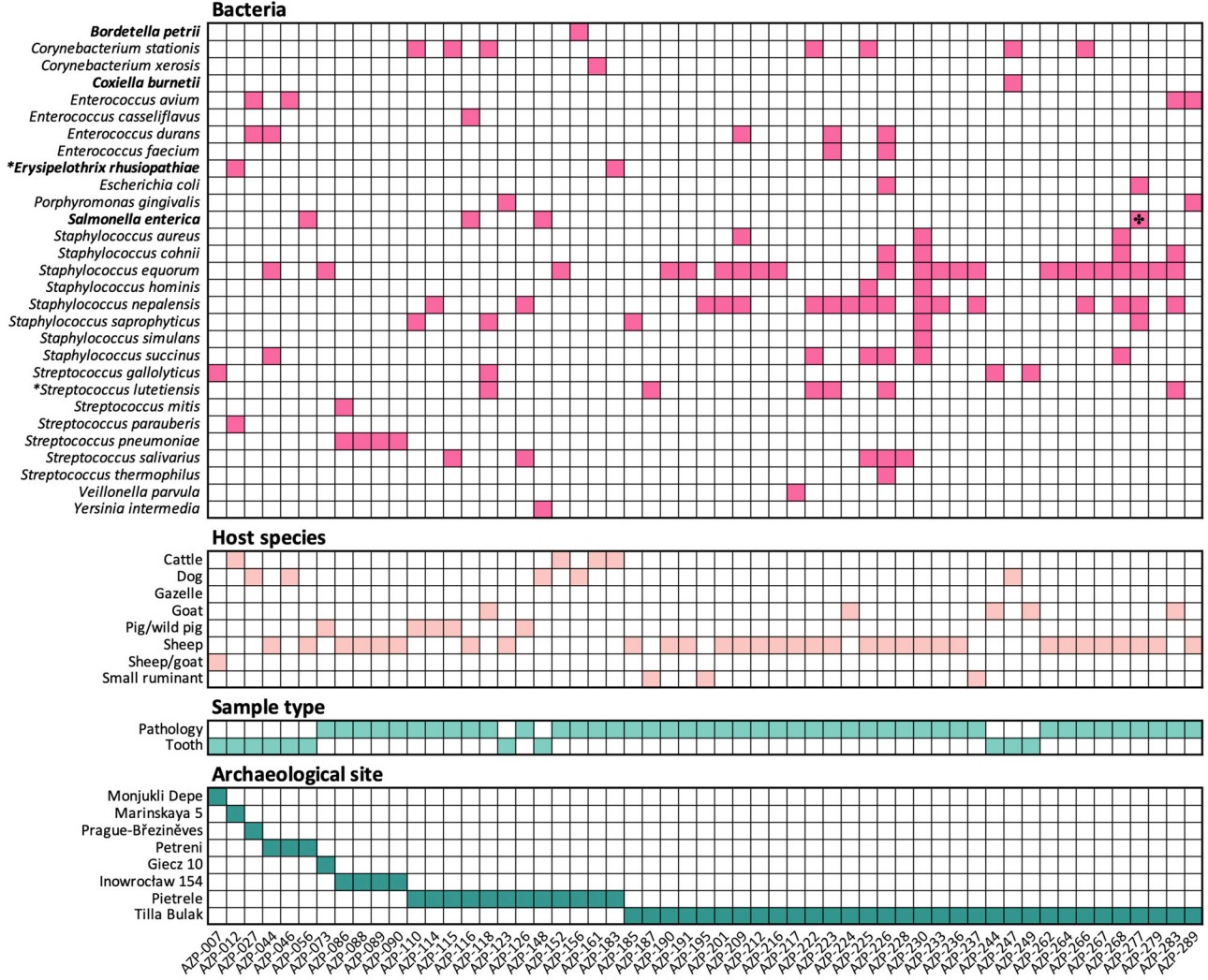

**Fig. 4 | Host, sample type, and archaeological site information for all identified bacterial pathogens.** Strictly pathogenic bacterial species in humans shown in bold. * Marks species with sufficient data for phylogenetic placement. ✤ Denotes the sample, which passed our stringent quality criteria at the taxonomic id of *Salmonella enterica* subsp. *enterica* as well as the ancestral taxonomic node of *S. enterica*.

$p < 0.0001$, respectively; Fisher Exact test; Supplementary Data 2). This suggests that DNA preservation may have been overall better at the Bronze Age sites Tilla Bulak and Inowrocław, especially compared to the samples from the much older Pietrele site (4550–4250 BCE), which were also studied for pathology, processed similarly in the laboratory, composed 23% of the dataset, and yielded 22.2% of the bacterial hits. While age, deposition, and animal husbandry as well as sequencing depth differed across sites, it cannot be excluded that there was a higher pathogenic pressure during the period of deposition at Tilla Bulak compared to the time periods and geographical locations of the other sampled sites.

In summary, despite being mostly unspecific about the precise bacterial species, palaeopathological lesions can guide the prioritisation of promising bone specimens for the genomic investigation of pathogens. Moreover, our results suggest that geographic and presumably site-specific differences in bacterial DNA preservation exist, which may be driven by variation in the treatment of animal remains impacting DNA preservation.

### Phylogenetic placement of zooarchaeological pathogens

To understand the evolutionary relationship of the identified bacteria and further confirm their ancient authenticity, we aimed to estimate their phylogenetic placement. While the low coverage impeded a complete genome reconstruction, we instead placed ancient genomes with sufficient amounts of reads (>0.05X) into the genetic diversity of modern representatives of that same species. Therefore, we ascertained mutations among contemporary or high-quality ancient genomes of the bacterial species together with an outgroup. For each position identified as variable among the high-quality contemporary and ancient genomes, we set the base call for the ancient genome when all reads support a single base call (see Methods). While this approach does not allow us to infer the molecular evolution exclusive to the ancient genomes, it is informative about its phylogenetic placement to estimate the branching pattern and evolutionary relatedness to the modern variability[33].

Here, we explore the phylogenetic placement for two identified species that are zoonotic but primarily known to be pathogenic among livestock and for which we observed sufficient reads: *Erysipelothrix rhusiopathiae* and *Streptococcus lutetiensis*. First, *E. rhusiopathiae* is a multihost zoonotic pathogen with a significant economic impact particularly on swine as well as cattle production, but also as an occupational pathogen in humans[40]. We recovered an authentic ancient DNA signature in AZP-012 with over 19,000 reads (0.5X coverage, 38% breadth) being aligned to the reference genome 52683_D03 (Fig. S2,

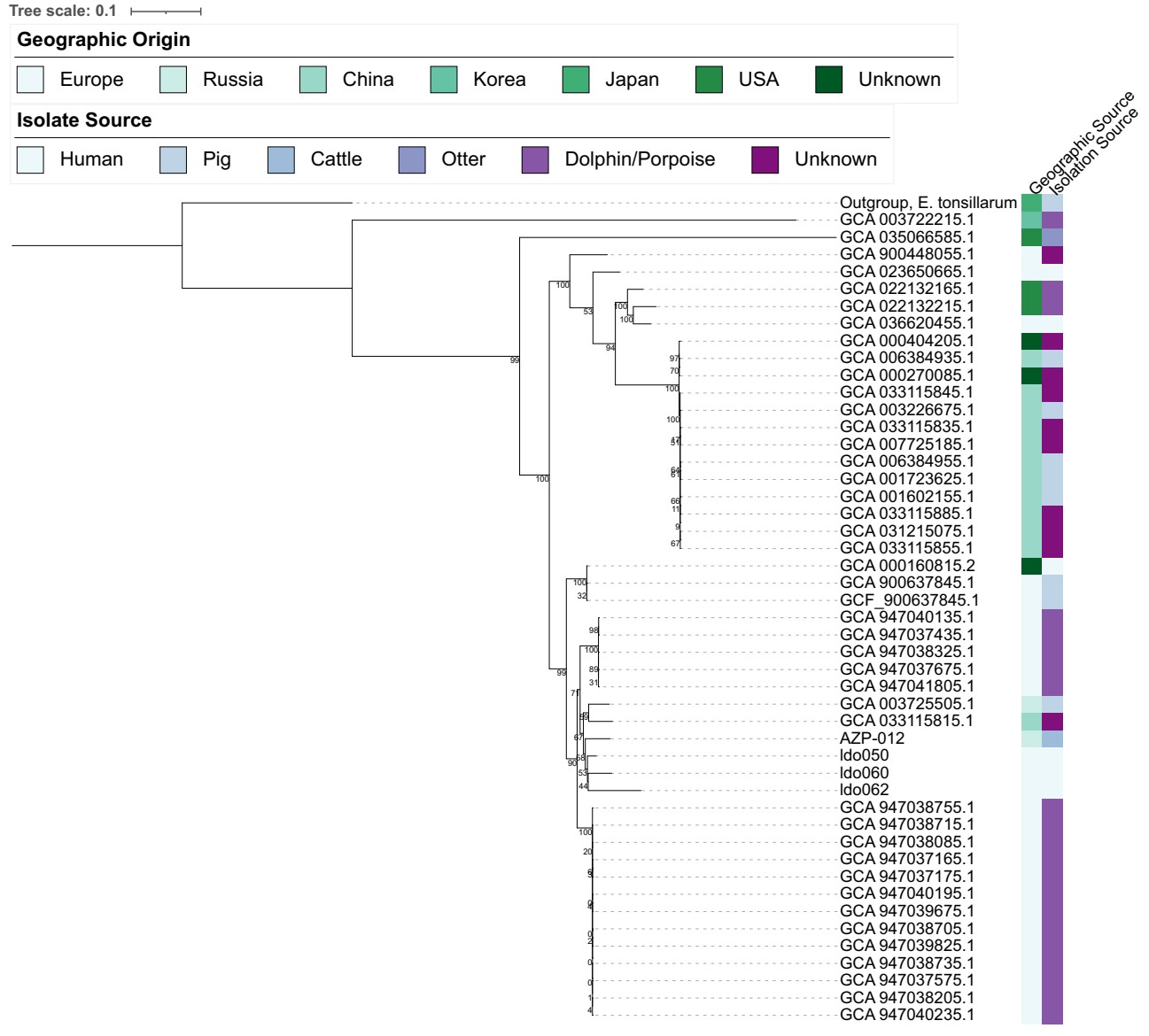

**Fig. 5 | Phylogenetic placement of AZP-12 within the *E. rhusiopathiae* diversity.**
Among 42 *E. rhusiopathiae* genomes, a human derived medieval genome[39] of
acceptable quality (>3x), and a single *E. tonsillarum* (outgroup) genome, we
ascertained 11,970 mutations. For the remaining 3 ancient genomes (AZP-012 [this
study] and two previously [Ido60, Ido62] published human-derived genomes[39]
passing quality control [≥0.05x]) the nucleotide call was assessed across these

positions (Supplementary Data 5), for placement into the phylogeny. AZP-012 from
Marinskaya 5 (North Caucasus) relates basal to the other ancient samples, as
expected based on the age of the samples. Felsenstein Bootstrap support (of 100
replicates) for each node is displayed. Transfer-Bootstrap-Expectation values are
shown in Fig. S4.

Supplementary Data 5). AZP-012 was sampled from a cattle tooth
excavated at Malinskaya 5 (Russia) dated to 2100–1800 BCE. We
identified 11,970 phylogenetically informative SNPs among 42 *E. rhu-
siopathiae* genomes isolated from a diverse set of contemporary hosts,
including domesticates and marine mammals, a published human-
derived medieval *E. rhusiopathiae* genome with sufficient coverage
(Ido050)[39] and the outgroup *E. tonsillarum* (Supplementary Data 4).
Assessing base calls across these positions on the identified low-
coverage ancient genome AZP-012 as well as two (Ido60 & Ido62)
additional, previously published medieval genomes with low coverage
(>0.05X)[39], the ancient genomes are phylogenetically placed at the
base of a sub-clade within *E. rhusiopathiae* (Fig. 5, Fig. S3). The boot-
strap value for the node separating the ancient samples from the clo-
sest modern genomes is only 67, driven by the resampling process and
the relatively few mutations (23) identified among all ancient samples
separating the ancient clade from the modern samples. However, the

Transfer-Bootstrap-Expectation[47] value of this node is high (0.952,
Fig. S4), indicating that the overall placement of the ancient genomes
is consistent, even though the exact reconstruction of the ancient
sample placement is unstable. Nevertheless, the phylogenetic place-
ment of AZP-012 together with independently generated ancient
genomes from Iberian human remains suggests *E. rhusiopathiae* was a
widespread multi-host pathogen in the past, harbouring within-species
diversity not yet described among modern representatives.

*S. lutetiensis* is a member of the *Streptococcus bovis* type sub-
group, along with the species *S. equinus*, *S. gallolyticus*, *S. infantarius*,
and *S. salivarius*[48]. *S. lutetiensis* has been primarily described to cause
bovine mastitis, a costly disease impacting dairy cattle worldwide, but
has also been isolated from other domesticates and humans[49,50]. After
alignment to the reference genome 45473_D02 we identified authentic
*S. lutetiensis* signatures in three samples: sheep AZP-223 (0.18X cov-
erage, 6.4% breadth), sheep AZP-226 (1.5X coverage, 70% breadth), and

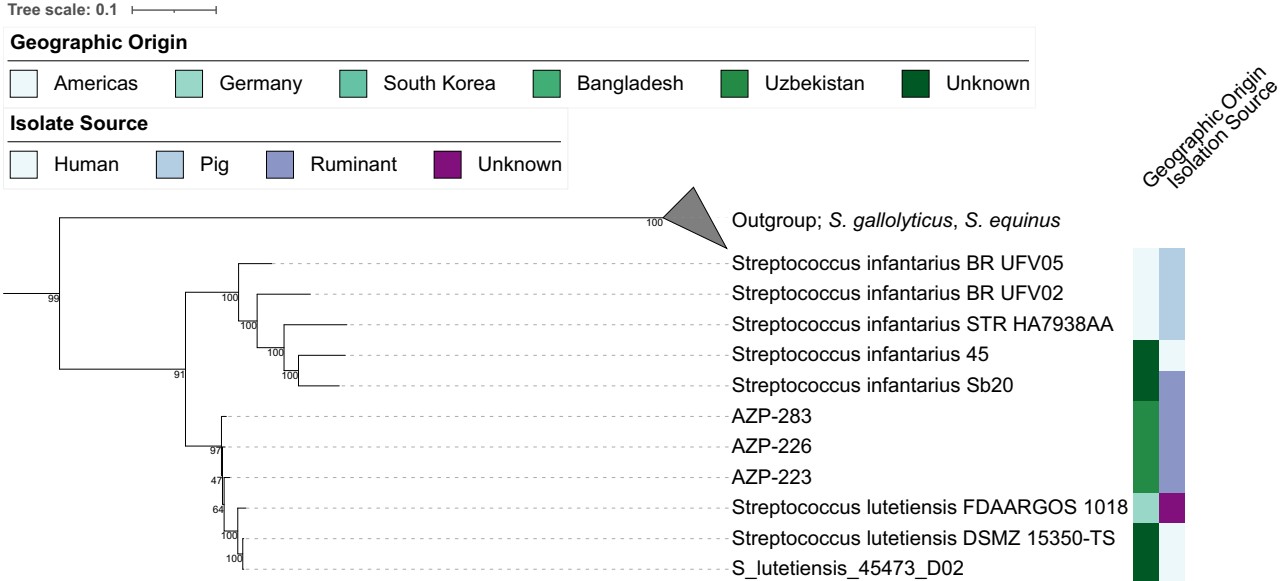

**Fig. 6 | Phylogenetic placement of AZP-283, AZP-226, and AZP-223 within the *Streptococcus* bovis-type subgroup.** Ancient genomes were placed within a phylogeny defined by 92,526 mutations, ascertained using in total 19 *S. lutetiensis, S. infantarius, S. equinus* and *S. gallolyticus* genomes that passed quality control (reference genome *S. lutetiensis* 45473_D02 (Supplementary Data 4, Supplementary Data 5). Collapsed outgroup clades include *S. gallolyticus* and *S. equinus*. All three ancient genomes originate from the Central Asian site of Tilla Bulak and form a monophyletic group basal to the known diversity of *S. lutetiensis*. Felsenstein bootstrap values are shown, the Transfer-Bootstrap-Expectation is presented in Fig. S5.

goat AZP-283 (0.11X coverage, 7% breadth), all excavated at Tilla Bulak (Uzbekistan) dated to 2000–1600 BCE (Fig. S2). In order to place the ancient genomic information into the modern genetic diversity, we leveraged 20 public genomes of *S. lutetiensis, S. infantarius, S. equinus, S. salivarius* and *S. gallolyticus* isolated from different sources across Eurasia. All modern genomes except the *S. salivarius* representative passed QC after alignment to the *S. lutetiensis* reference genome (Methods) (Supplementary Data 4). We identified 92,526 phylogenetically informative positions across the modern representatives. All three ancient genomes are placed monophyletic, basal to the *S. lutetiensis* diversity (Fig. 6, Supplementary Data 5). Again, the low number of known mutations, 23, identified across all ancient samples separating the three ancient genomes from the modern diversity leads to instability measured by the classic Felsenstein bootstrap (97 - 47); however, their overall placement basal to modern *S. lutetiensis* is stable using the Transfer-Bootstrap-Expectation (0.97, Fig. S5) and single-ancient-genome projections are placed consistently in 100% of Felsenstein bootstrap realisations (Fig. S6).

Taken together, the basal phylogenetic placement of *S. lutetiensis* and the clustering of the ancient genomes from *E. rhusiopathiae* further corroborates the ancient authenticity of the identified pathogen DNA and provides evidence for the widespread distribution and deep evolutionary history of both pathogens with domesticated animals. Nevertheless, the analyses of these pathogens are restricted to a few host species due to the limited representation in available databases, which prevented the phylogenetic projection of other candidates, like *Staphylococcus nepalensis*. As such, our inference is only a first step into the largely uncharted territory of prehistoric infectious disease in animals and livestock, and calls for further investigation of zoonotic pathogens both through the zooarchaeological record and with increased surveillance in modern contexts.

## Discussion

Prehistoric disease reservoirs in faunal populations and the likely increase in pathogen exposure that followed animal domestication are poorly understood and challenging to investigate. In this study, we combined palaeopathological analysis and ancient genomics to explore whether a selective sampling approach focused on palaeo-pathological lesions indicative of infection or teeth provide a useful starting point for ancient DNA sampling for faunal palaeomicrobiological analysis.

While we recovered ancient pathogen DNA from both bones with palaeopathological lesions and teeth, our results suggest that leveraging samples with pathologies offers advantages over random tooth sampling and provides a focused and economic method for selecting samples in pathogen DNA retrieval from zooarchaeological remains. Nevertheless, targeting palaeopathological lesions for pathogen DNA retrieval has several limitations. The lesion itself may not be the optimal sampling location for some pathogens, but the absence of a certain diagnosis inhibits the customisation of a DNA sampling strategy based on the pathophysiology of the pathogen. Lesions are also likely to be less optimal than dense bone for long-term DNA preservation due to the processes of bone remodelling and degradation. In addition, pretreatment methods designed to reduce contamination from environmental sources or human handling may have an adverse effect on pathogen DNA recovery, as UV radiation risks destroying pathogen DNA in lesions where bone density is low, while bleach washing or predigestion may release pathogen DNA that is not bound deeply within the extracellular matrix[51]. Moreover, we observe variation in DNA preservation across sites presumably due to differences in depositional practices and environmental conditions, but potentially also influenced by specimen handling and storage, all of which may contribute to the differences observed in recovered pathogen DNA across sites. In addition, different methods for DNA laboratory processing as well as sequencing depth might influence pathogen DNA recovery; in this study, the only site, Tilla Bulak, where the number of robust pathogen hits rose to statistically significant levels was processed using the single-stranded Santa Cruz protocol[52] for library preparation. Future studies explicitly designed with comparable amounts of pathological and non-pathological specimens per site as well as further investigation of the optimal DNA laboratory processing strategies will be instrumental to further corroborate our findings.

The majority of palaeopathological lesions are the result of long term and chronic infections as bone remodelling is generally a slow

process[53]. Although some infections, like acute osteomyelitis, can rapidly induce skeletal changes[54–56], individuals who perished from acute infections are often not visibly identifiable in the archaeological record[57,58]. It is also worth noting that identified bacterial hits often cannot be conclusively linked to the observed lesions, which tend to be non-specific. The bacteria may also represent opportunistic colonisation of an existing site of infection or they could have been transferred to the sample in the depositional environment from another infected source. Although we recovered a greater proportion of robust bacterial hits from palaeopathological lesions, we do not recommend excluding teeth when designing a sampling strategy for future faunal palaeomicrobiology studies. Many ancient bloodborne pathogens that entered the bloodstream as a result of bacteraemia have been recovered from the dental pulp chambers of humans without any osteological evidence of disease[23,28]. However, it is important to bear in mind that tooth morphology differs between species; for example, humans, dogs, and pigs (except the pig tusks/canines) have brachydont teeth with enclosed pulp chambers, while sheep, goats, and cattle have hypsodont teeth, which do not have enclosed pulp chambers and often have open roots. While we do not observe a statistically significant difference in the number of robust bacterial hits obtained from these two types of teeth, more research is needed to understand how tooth morphology affects the preservation and recovery of authentic ancient pathogen DNA.

Our dataset yielded several bacterial hits including four *S. enterica* signatures, a group of bacteria responsible for gastroenteritis or systemic disease in humans and animals[34,35,59], from three sites across Eurasia covering a period between 4000 and 8000 years ago. Interestingly, two of the four signatures were from teeth, one was from a pathological mandible, and one was from a lesion on a femur, pointing at additional skeletal material that may serve as a source for ancient *S. enterica* DNA. The emergence of human salmonellosis has previously been linked to the Neolithisation of Eurasia and likely has a zoonotic origin[23]. Although the screening dataset did not provide sufficient genome coverage for further analysis of this bacterium, future deep sequencing or target enrichment capture may provide more clarity on the validity and relation between these animal-derived hits and known ancient and modern *S. enterica* diversity. Similarly, we did not obtain sufficient genome coverage to further investigate the two sheep-derived hits to *P. gingivalis*, a commensal oral bacterium that is also a member of the polymicrobial 'red complex' associated with periodontal disease in humans[43] and sheep[44]. Although this bacterium is most frequently recovered from the oral cavity, one of our hits originated from a palaeopathological lesion on a femur, which may represent a case of localised bone infection, although contamination cannot be ruled out. Previous research revealed a shift in the composition of the oral microbiota following the Neolithic Transition with agriculturalist populations having a greater number of taxa associated with periodontal disease, including *P. gingivalis*[60]. In the future, animal-derived *P. gingivalis* genomes may help clarify the history of periodontal disease among human-associated livestock as well as potential zoonotic and reverse zoonotic spillovers.

We also detected other known zoonotic pathogens, where *C. burnetii*[37] and *E. rhusiopathiae*[38] are widespread occupational pathogens today. The identification of several human pathogens from the zooarchaeological record highlight the potential for zoonotic and reverse-zoonotic disease emergence as a risk factor during the Neolithisation across Eurasia. Consistent with this hypothesis, Sikora et al.[27] identified an uptick in zoonotic pathogens in particular among pastoralist communities, highlighting the risks of our close relationships with domesticates. Indeed, the reconstruction of a *Y. pestis* genome from a Bronze Age sheep points at domesticated animals as bridge hosts linking the pathogen reservoir with humans[14], providing further credence to the hypothesis. Although most of the authenticated bacterial genomes in this study had too low coverage for further analysis,

the four samples representing two species used in the phylogenetic analyses suggest that the authentication process is robust. While such analyses were not possible for all potential pathogens detected here, they are important in order to distinguish between signatures of authentic endogenous pathogens. Especially opportunistic pathogens present in the human microbiome pose a risk, as they may have been deposited during excavation and subsequent handling, and could have developed degradation patterns reminiscent of authentic ancient DNA while in storage.

Ancient DNA offers a powerful means of detecting infectious agents in animals. Although this study demonstrates the benefit of selectively sampling pathological bones for palaeomicrobiology, ancient DNA sequencing also provides a unique opportunity to identify infections in skeletal elements lacking visible evidence of disease and those in which the evidence is obscured by taphonomic changes. By enabling the recovery of pathogen DNA from these remains, ancient DNA analysis provides a more comprehensive picture of past zoonotic and reverse zoonotic infections and their evolutionary trajectories, helping to uncover the long-term dynamics of host-pathogen interactions and the role of animals in disease emergence and spread. Hence, this study is an important step towards exploring ancient pathogens in the zooarchaeological record, necessary to expand the One Health concept into the past, which holds the promise to elucidate the reservoir and host range of zoonotic pathogens, the geographic and temporal spread, and the genetic mechanisms enabling the evolutionary adaptation towards the human host.

## Methods

### Sites, faunal assemblages, and palaeopathological analysis

Zooarchaeological assessments of the faunal assemblages had been carried out for all sites prior to this study. All sites are listed in Supplementary Data 1 and a summary of the sites can be found in the Supplemental Online Material. Apart from availability, the choice of sites was dictated by several factors. First, each of the sites is set in a different environmental context, and this variety provided the opportunity to avoid biases due to the environment on DNA and palaeopathology preservation potentially present at single sites. Second, their different chronologies may capture pathogen emergence within animal populations over time. Third, the geographic range of sites with samples positive for ancient pathogen DNA can help indicate the range of past zoonotic disease.

Altogether, the study materials consisted of 346 specimens. Detailed archaeological, palaeopathological, and metagenomic information for each sample can be found in Supplementary Data 2. Among the study materials were 188 bones with pathological lesions that were selected following palaeopathological analysis. The analysis was based on palaeopathological literature[19,21,53,61–65] and comparative collections stored in the Institute of Geology at Adam Mickiewicz University, Poznań and German Archaeological Institute in Berlin. Cases of pathology were classified according to Bartosiewicz and Gal[66]. The recording system for cases of pathology followed the protocol of Vann and Thomas[67], and included recording all cases of lesions, without focusing on extreme or rare ones. All locations of the palaeopathological lesions presented in Supplementary Data 2 were recorded according to the Nomina Anatomica Veterinaria[68]. To explore the possibility of preservation of DNA from blood-borne pathogens in non-human animal teeth, we also included 131 teeth in the study. These came from all sites except Ipatovo 3 (Russia), Ransyrt 2 (Russia), Bürgermeister-Ulrich-Straße 100 (Germany), Universitätsstraße (Germany), and the sites in Poland.

No selection was made based on wild or domesticated animals, but given the post Neolithic ages of the contexts from which all specimens originated, the abundance of skeletal elements from domesticated species outweighed those from wild species. We also investigated the taphonomic condition of the specimens, considering

the impact of factors such as weathering, trampling, gnawing, butchering, manufacturing, water, plant roots, and crystallisation of minerals[69–72], to check environmental factors that might have affected the preservation of any pathological lesion. The conclusion is that all specimens with pathological lesions used here were not affected by post-depositional processes that could have an impact on the appearance and extent of any pathological lesion.

## Lab processing

**Drilling.** All samples were processed in dedicated clean room facilities. Aside from the German samples, which were processed entirely at the ancient DNA facility in Tartu, Estonia, all selected samples were drilled in the ancient DNA laboratory at the MPIIB in Berlin. With the exception of the German samples, only one sample was collected from each element. For some of the German specimens multiple subsamples were collected from bones, and when mandibles or maxillae included teeth, these were also sampled. Occasionally, multiple teeth were collected from the same individual and analysed as separate samples (Supplementary Data 2).

For tooth sampling, the roots were sawed off and powder was collected by drilling into the root, the pulp cavity, or both depending on the morphology and size of the tooth. Notably, pig teeth had very fragile roots that could not be drilled without fragmenting. Due to their size, small/ milk teeth were wiped with bleach solution, cleaned with water, and then pulverised entirely using a mortar and pestle. The German teeth were not drilled; instead, a tooth root was collected as a chunk sample.

Palaeopathological lesions were sampled directly. No pretreatment of bone was applied in order to prevent potential destruction of any pathogen DNA present in the lesions. For each lesion, drilling was carried out across the surface of the lesion and, where possible, this was complemented with an additional sampling by drilling deeper into the cancellous bone beneath the lesion. When possible, trabecular bone was also collected. Bones without lesions were sampled in a similar manner to those with palaeopathological lesions. Bone fragments were collected as chunk samples from AZP-316 and AZP-318.

Approximately 50 mg bone or tooth powder was collected from each sample. In cases where larger amounts of powder had been collected, samples were pulse vortexed three times and then 50 mg of the mixed powder was weighed out.

**DNA extraction, libraries, and sequencing.** Tartu clean room: samples from the Czech Republic, Germany, Moldova, Russia, Turkmenistan, and the Polish sites Giecz 10 (with the exception of AZP-080), Gdańsk, Krzczonowice, and Zdrojówka were either taken as chunk samples or drill powder without any decontamination or pretreatment. DNA extraction was performed following the EDTA-Proteinase K protocol by Dabney et al.[73] according to Keller and Scheib[74,75] with the modification of incubation at 37 °C instead of room temperature and incubation for only 24 h for drill powder. Chunk samples were incubated for 72 h. Double-stranded, non-UDG-treated libraries were generated following the Meyer and Kircher protocol for double-stranded DNA[76] followed by unique dual indexing PCR for 14 cycles according to Keller et al.[77,78] with the following modifications: in the indexing PCR, 10 µl of BSA (20 mg/ml) was replaced with water to reduce the risk of contamination with bovine DNA; 8 µl (10 µM) of NEBNext Multiplex Oligos for Illumina (sets E6440 and E6442) were used instead of a unique pair of 4 µl (10 µM) of i7 and i5 indexing primers each. After indexing, DNA concentrations of libraries were measured with a Qubit® dsDNA HS Assay Kit on a Qubit® Fluorometer. The average fragment length per library was determined using the Agilent Technologies 2200 TapeStation system with either regular or High Sensitivity D1000 screen tape. After amplification and pooling, the DNA libraries were sequenced Illumina NextSeq 500 platform with the HIGH 150 cycle kit in paired-end (PE) mode.

Berlin clean room: DNA extraction was carried out on the samples from Pietrele, Tilla Bulak and the Polish sites Inowrocław 154, Izdebno Kościelne, Moszna Wieś, and sample AZP-080 from Giecz 10 using methods optimised for ancient DNA[73,79]. 1 mL of lysis buffer consisting of 5 M EDTA (pH 8.0), Tween 20, Proteinase K, and $H_2O$ was added to each 50 mg bone or tooth sample. The tubes were then sealed with parafilm and incubated with rotation overnight at 37 °C. Following lysis, samples were centrifuged for 2 Min at $16,400\,g$ to pellet any undissolved material. The supernatant was collected and transferred to 10 mL of binding buffer (5 M guanidine hydrochloride, 40% (vol/vol) 2-propanol, 0.12 M sodium acetate, and 0.05% (vol/vol) Tween 20). Samples were then filtered through minElute columns and washed twice with 200 µL PE buffer (Qiagen) before elution in 50 µL EBT. Single-strand DNA libraries were constructed following the Santa Cruz Reaction protocol[52]. A qPCR test was performed to determine the optimal number of PCR cycles for each sample using Maxima SYBR Green/ROX qPCR Master Mix (Thermo Scientific) following the set-up recommended by the manufacturer on a OneStepPlus machine (Applied Biosystems). Unique dual indexing PCR was performed using the AmpliTaq Gold 360 Mastermix (Applied Biosystems). PCR products were purified using a 1.2 ratio of Sera-mag speed beads (Fisher Scientific). Beads were washed twice with 200 µL ethanol (80%) and eluted in 30 µL EBT. DNA concentrations were measured using the Qubit 1X dsDNA High Sensitivity kit (Invitrogen), and average fragment lengths were measured on a Fragment Analyzer (Agilent) using the High Sensitivity NGS Fragment kit (Agilent). Samples were pooled based on equimolarity and sequenced on Novaseq SP flowcells in 50PE mode. Extracts and libraries can be made available upon request for further investigations.

## Bioinformatic processing

**Screening pipeline.** nf-core/eager was used for bioinformatic processing, including quality control, initial mapping to the human reference genome, and finally metagenomic screening of unmapped reads for candidate pathogens using MALT and HOPS[30,31]. See Raw Data Processing in project github for parameters. A list of taxa of candidate pathogen species and genera can be found in (Supplementary Data 3); this includes the HOPS screening list of known zoonotic pathogens and genera, and was extended following a literature search with additional zoonotic and animal-specific pathogens. Note that the species list is not exhaustive. We removed the genus node of *Bacillus*, *Clostridium* and *Brucella* due to closely related non-pathogenic species highly abundant in soil that may result in false positive signatures. Following this, we checked all entries for overlap with the NCBI taxonomy database and removed non-findable names. We removed any duplicate entries from the HOPS list. A custom screening database was constructed based upon all bacteria and archaea available from RefSeq[80] that includes only the minimal set of genomes per taxonomic ID to represent the microbial intra-species diversity and minimise database size[14] (see github for list of genomes included in the database *pathogen_screening/database_info/all_metadata_database_genomes.tsv*). A custom HOPS-post-processing script was used to identify candidate samples, which allows for the manual setting of various heuristic thresholds and output was limited to samples with pathogen hits that fulfil all of our set criteria. The criteria were: (1) a minimum number of 50 reads assigned to a taxonomic node of interest, (2) a default edit distance ratio of 0.8, (3) an ancient edit distance ratio of 0.5, and (4) a minimum read distribution of 0.6, and (5) damage on 10% of reads with possible damage transition patterns (C → T) at the 5′ end; or, also the 3′ end (G → A), in the case of double stranded libraries. The read distribution metric (number of covered genome positions/sum of aligned read lengths)[30] is designed to be highly attuned to detecting particularly weak pathogen signatures, since the value decreases with any read stacking which would be expected from spurious misalignments by reads from organisms with homologous genome segments

within the metagenomic sequencing library, a major concern for false positives. However, this metric is inappropriate for high values of reads assigned to a specific candidate pathogen node, since with many reads, we expect some measure of overlap, leading to a reduced read distribution value. To ensure that we identified any pathogen signatures constituted from many reads for which the read distribution cutoff would be inappropriate, we separately ran the post-processing script with the same requirements, except for minimum number of reads >10,000 and no read distribution cutoff.

**Host validation with kraken2.** The taxonomic profiling pipeline nf-core/taxprofiler was used to confirm, identify, or correct the host identity of each metagenomic dataset[81]. We performed short-read sequencing quality control (--perform_shortread_qc) and utilised kraken2 with a prebuilt nt database dated 11/29/2023 retrieved from https://benlangmead.github.io/aws-indexes/k2 to taxonomically assign all reads that passed QC using the --kraken2_save_minimzers option. Metagenomic host classification can be challenging due to high variability in sample preservation and contamination with environmental DNA, both of which interfere with reliable taxonomic host identification. Moreover, several eukaryotic species are regularly identified in ancient metagenomic datasets, for example European hedgehog (*Erinaceus europaeus*), suggesting contamination also at reference genome level leading to spurious results[82]. To identify the host genus, we identified the top genus taxonomic nodes below the Metazoa kingdom node and set two cutoffs for assessing the host identity using metagenomic taxonomic classification. For samples with multiple libraries generated from the same extract, we assessed the library with the highest proportion of reads assigned to the genus level (range of assessed QC passed reads per sample: 1,147,688-45,286,790). First, we set a minimum threshold of 0.1% of input QC passed reads assigned to a given taxonomic node for it to be trusted as a host identification when the host genus was in agreement with the prior morphological identification, when available. A total of 39.4% of samples reached that threshold of which 77.9% agreed with the morphological host identification. Values below 0.1% of a single genus assignment indicate poor sample preservation at the endogenous DNA level and were not considered further for taxonomic identification. Second. a threshold of 1% of reads was required to update a host identity in the case that the morphological host identity was inconsistent with the metagenomically-assigned top genus, unless the metagenomically-assigned top genus was *Homo*, which presumably stems from contamination. Only 14.6% of samples reached that threshold, of which 17.3% of the sample's host species was updated. When no morphological host identification was possible, or multiple potential hosts were proposed, the 0.1% threshold was used for metagenomic host genus assignment. Results are shown in Supplementary Data 2. No correlation between host DNA and pathogen DNA preservation was observed (Fig. S1).

**Pathogen phylogenetic reconstruction.** To validate our metagenomic microbial screening results and learn about the evolutionary history of the identified ancient pathogens, we performed phylogenetic projection of the ancient genomes into the known modern diversity of the respective species. We focused on two known pathogenic species - *E. rhusiopathiae* and *S. lutetiensis* - each with at least one ancient sample with a high number of reads (3402–7476) assigned specifically to the species node during metagenomic screening, which present the greatest likelihood of having enough genetic data for phylogenetic placement. Within our dataset we identified signatures for two *E. rhusiopathiae* and six *S. lutetiensis* candidates (Fig. 4), which we explored for sufficient quality for phylogenetic placement (Supplementary Data 5). Due to the low coverage expected for genomes reconstructed from shotgun sequencing, we do not call SNPs directly in these samples unless they exceed a coverage of 3X. Instead, we first identify phylogenetically informative SNPs from high-quality modern and ancient genome comparisons, and then assess support for base calls on these positions in the ancient genomes. This approach minimises the possibility of erroneous base calls due to metagenomic mismappings or ancient DNA damage, since only positions already known to define the differences between major branches of diversity in the modern genomes are assessed.

For *E. rhusiopathiae*, we utilised all RefSeq genome assemblies available in August 2024 and additional representatives from *E. tonsillarum*, *E. inopinata* and *B. extructa* as outgroups, in line with a previous investigation of the ancient and modern diversity of the pathogen[39] (*N* = 50, including reference genome). For *S. lutetiensis*, we utilised an ad hoc selection of genomes with raw sequencing data available from ENA based on the species and related species in the *S. bovis* subgroup including *S. infantarius*, *S. salivarius*, *S. gallolyticus*, and *S. equinus*, aiming to include a representative sampling of the geographic and host diversity in the species, despite inconsistencies in the reporting of this information across various sequencing records (*N* = 21, including reference genome) (Supplementary Data 4). For both *E. rhusiopathiae* and *S. lutetiensis*, the listed genomes were included to optimise the genetic representation due to the overall limited number of available genomes and despite the inconsistencies in available metadata. Genomes from GenBank without any available raw sequencing reads were used as input for wgsim (https://github.com/lh3/wgsim) to simulate sequencing data to remap the samples to the reference genome. Additionally, for *E. rhusiopathiae*, we reanalysed seven additional publicly available ancient metagenomes previously identified as positive for *Erysipelothrix* genus from Iberia from the past 1000 years[39].

First, we generated deduplicated bams for ancient and modern genomes using nf-core/eager (see Raw Data Processing in project github for parameters for ancient and modern samples). We masked three terminal bases in the ancient samples known to be affected by ancient DNA damage. Then, all deduplicated bam whose average genome coverage after mapping quality filtering of MQ30 surpassed 0.05X were input into an in-house Snakemake variant calling pipeline utilising samtools which collects support statistics for variant positions across all samples for downstream processing using an interactive python environment[83] (see also data availability statement).

Within the interactive python environment, we performed stringent quality control to identify phylogenetically informative mutations present among the genomes of sufficient quality. We utilised the following quality control thresholds: First, we only considered mapped genomes with a global coverage above 3X for the initial identification of phylogenetically informative mutations. For *E. rhusiopathiae*, this removed 14 genomes, four *E. rhusiopathiae* ingroup genomes which are suppressed in the NCBI database due to failed taxonomy match checks as of 10/2025 (Supplementary Data 4), and the outgroup genomes of *E. inopinata* and *B. extructa*, and eight ancient genomes which fell below 3x coverage. For *S. lutetiensis*, this removed the outgroup *S. salivarius* genome. Second, the nucleotide call in any modern sample was set to N (uncalled base) if it failed any of the following criteria: the FQ score was greater than -30, the major allele frequency was below 90%, fewer than 2 forward reads supported the position, fewer than 2 reverse reads supported the position, or more than 50% of reads supporting a base call also supported an indel call within 3 bases. Lastly, we subset the positions for downstream analysis to only the core genome, which we defined as positions called (not set to N) with the above thresholds across 100% of retained public genomes.

To generate loose base calls on our ancient metagenomic genomes, we took the following approach: Utilising only the phylogenetically informative positions (variable SNPs) that we identified above in the public diversity, we assessed base-calls in the low coverage ancient genomes presented here, and retained samples with an average coverage across unfiltered variant positions of >0.05X after

filtering reads above Q30 for phylogenetic placement (removing a single ancient *E. rhusiopathiae* candidate and three ancient *S. lutetiensis* candidates [Supplementary Data 4]). We called bases on these positions if the position passed the following quality cutoffs: coverage of ≥1, FQ score lower than -30 and major allele frequency equal to 1; otherwise the genotype was set to "N". The final SNP tables utilised for the phylogenetic analysis below can be found in the github for the project and are summarised in Supplementary Data 5.

Using the SNP table, we generated a maximum likelihood (ML) tree using RAxML-ng[84] with 100 bootstrap replicates. To further validate the phylogenetic placement of the ancient samples in the ML tree, we undertook two additional analyses. First, we calculated the Transfer-Bootstrap-Expectation metric using RAxML-ng using 100 bootstrap replicates. This metric is useful for assessing the overall stability of a node relative to the rest of the phylogenetic tree. Second, to ensure that the high level of missingness in ancient samples (range 23–90%) was not biasing their phylogenetic placement, we created a ML tree with 100 bootstrap replicates separately for each ancient sample with a SNP table subset to sites called in that ancient sample. The topology of the ML realisation trees broadly recaptures the tree topology observed when allowing for missing data, and are consistent across different ancient genomes (Fig. S3 and S6). Phylogenetic trees were visualised using iTOL and were annotated for isolation source and geographic source using publicly available metadata.

## Statistical tests

Fisher Exact tests were used for comparisons of preservation by site and pathogen recovery by site by comparing the number of samples from that site with or without >1% host DNA assigned or a pathogen identification compared to all other sites collapsed, respectively. Multiple testing correction was done by the Bonferroni method (adjusted alpha = 0.00189). Only p-values significant after multiple testing corrections have been reported. Comparisons for palaeopathological lesions for given sites and for recovery of pathogen hits were conducted only on samples from sites that were assessed for palaeopathologies (Supplementary Data 1).

## Reporting summary

Further information on research design is available in the Nature Portfolio Reporting Summary linked to this article.

# Data availability

Source data are provided with this paper. DNA sequencing data generated in this study is available at the European Nucleotide Archive under project accession PRJEB63473. Individual accession identifiers are also listed in Supplementary Data 2. We reanalysed existing datasets for the generation of phylogenies: PRJEB72246 [https://www.ebi.ac.uk/ena/browser/view/PRJEB72246] GCF_900637845.1 [https://www.ncbi.nlm.nih.gov/datasets/genome/GCF_900637845.1] GCF_000373785.1 [https://www.ncbi.nlm.nih.gov/datasets/genome/GCF_000373785.1] GCF_014396165.1 [https://www.ncbi.nlm.nih.gov/datasets/genome/GCF_014396165.1] GCF_000177375.1 [https://www.ncbi.nlm.nih.gov/datasets/genome/GCF_000177375.1] GCA_000160815.2 [https://www.ncbi.nlm.nih.gov/datasets/genome/GCA_000160815.2] GCA_000270085.1 [https://www.ncbi.nlm.nih.gov/datasets/genome/GCA_000270085.1] GCA_000404205.1 [https://www.ncbi.nlm.nih.gov/datasets/genome/GCA_000404205.1] GCA_001602155.1 [https://www.ncbi.nlm.nih.gov/datasets/genome/GCA_001602155.1] GCA_001723625.1 [https://www.ncbi.nlm.nih.gov/datasets/genome/GCA_001723625.1] GCA_003226675.1 [https://www.ncbi.nlm.nih.gov/datasets/genome/GCA_003226675.1]

GCA_003722215.1 [https://www.ncbi.nlm.nih.gov/datasets/genome/GCA_003722215.1]
GCA_003725505.1 [https://www.ncbi.nlm.nih.gov/datasets/genome/GCA_003725505.1]
GCA_006384935.1 [https://www.ncbi.nlm.nih.gov/datasets/genome/GCA_006384935.1]
GCA_006384955.1 [https://www.ncbi.nlm.nih.gov/datasets/genome/GCA_006384955.1]
GCA_007725185.1 [https://www.ncbi.nlm.nih.gov/datasets/genome/GCA_007725185.1]
GCA_022132165.1 [https://www.ncbi.nlm.nih.gov/datasets/genome/GCA_022132165.1]
GCA_022132215.1 [https://www.ncbi.nlm.nih.gov/datasets/genome/GCA_022132215.1]
GCA_023650665.1 [https://www.ncbi.nlm.nih.gov/datasets/genome/GCA_023650665.1]
GCA_031215075.1 [https://www.ncbi.nlm.nih.gov/datasets/genome/GCA_031215075.1]
GCA_033115815.1 [https://www.ncbi.nlm.nih.gov/datasets/genome/GCA_033115815.1]
GCA_033115835.1 [https://www.ncbi.nlm.nih.gov/datasets/genome/GCA_033115835.1]
GCA_033115845.1 [https://www.ncbi.nlm.nih.gov/datasets/genome/GCA_033115845.1]
GCA_033115855.1 [https://www.ncbi.nlm.nih.gov/datasets/genome/GCA_033115855.1]
GCA_033115885.1 [https://www.ncbi.nlm.nih.gov/datasets/genome/GCA_033115885.1]
GCA_035066585.1 [https://www.ncbi.nlm.nih.gov/datasets/genome/GCA_035066585.1]
GCA_036620455.1 [https://www.ncbi.nlm.nih.gov/datasets/genome/GCA_036620455.1]
GCA_900448055.1 [https://www.ncbi.nlm.nih.gov/datasets/genome/GCA_900448055.1]
GCA_900637845.1 [https://www.ncbi.nlm.nih.gov/datasets/genome/GCA_900637845.1]
GCA_902772725.1 [https://www.ncbi.nlm.nih.gov/datasets/genome/GCA_902772725.1]
GCA_902781835.1 [https://www.ncbi.nlm.nih.gov/datasets/genome/GCA_902781835.1]
GCA_902795695.1 [https://www.ncbi.nlm.nih.gov/datasets/genome/GCA_902795695.1]
GCA_902797585.1 [https://www.ncbi.nlm.nih.gov/datasets/genome/GCA_902797585.1]
GCA_947037165.1 [https://www.ncbi.nlm.nih.gov/datasets/genome/GCA_947037165.1]
GCA_947037175.1 [https://www.ncbi.nlm.nih.gov/datasets/genome/GCA_947037175.1]
GCA_947037435.1 [https://www.ncbi.nlm.nih.gov/datasets/genome/GCA_947037435.1]
GCA_947037575.1 [https://www.ncbi.nlm.nih.gov/datasets/genome/GCA_947037575.1]
GCA_947037675.1 [https://www.ncbi.nlm.nih.gov/datasets/genome/GCA_947037675.1]
GCA_947038085.1 [https://www.ncbi.nlm.nih.gov/datasets/genome/GCA_947038085.1]
GCA_947038205.1 [https://www.ncbi.nlm.nih.gov/datasets/genome/GCA_947038205.1]
GCA_947038325.1 [https://www.ncbi.nlm.nih.gov/datasets/genome/GCA_947038325.1]
GCA_947038705.1 [https://www.ncbi.nlm.nih.gov/datasets/genome/GCA_947038705.1]
GCA_947038715.1 [https://www.ncbi.nlm.nih.gov/datasets/genome/GCA_947038715.1]

GCA_947038735.1 [https://www.ncbi.nlm.nih.gov/datasets/genome/GCA_947038735.1]

GCA_947038755.1 [https://www.ncbi.nlm.nih.gov/datasets/genome/GCA_947038755.1]

GCA_947039675.1 [https://www.ncbi.nlm.nih.gov/datasets/genome/GCA_947039675.1]

GCA_947039825.1 [https://www.ncbi.nlm.nih.gov/datasets/genome/GCA_947039825.1]

GCA_947040135.1 [https://www.ncbi.nlm.nih.gov/datasets/genome/GCA_947040135.1]

GCA_947040195.1 [https://www.ncbi.nlm.nih.gov/datasets/genome/GCA_947040195.1]

GCA_947040235.1 [https://www.ncbi.nlm.nih.gov/datasets/genome/GCA_947040235.1]

GCA_947041805.1 [https://www.ncbi.nlm.nih.gov/datasets/genome/GCA_947041805.1]

GCF_900475675.1 [https://www.ncbi.nlm.nih.gov/datasets/genome/GCF_900475675.1]

SRX5992946 [https://www.ebi.ac.uk/ena/browser/view/SRX5992946]

SRX6387061 [https://www.ebi.ac.uk/ena/browser/view/SRX6387061]

SRX5764132 [https://www.ebi.ac.uk/ena/browser/view/SRX5764132]

SRX2155810 [https://www.ebi.ac.uk/ena/browser/view/SRX2155810]

SRX2155807 [https://www.ebi.ac.uk/ena/browser/view/SRX2155807]

SRX2155790 [https://www.ebi.ac.uk/ena/browser/view/SRX2155790]

SRX2122668 [https://www.ebi.ac.uk/ena/browser/view/SRX2122668]

SRX6477869 [https://www.ebi.ac.uk/ena/browser/view/SRX6477869]

SRX9293568 [https://www.ebi.ac.uk/ena/browser/view/SRX9293568]

SRX9689842 [https://www.ebi.ac.uk/ena/browser/view/SRX9689842]

SRX6477820 [https://www.ebi.ac.uk/ena/browser/view/SRX6477820]

ERX4964206 [https://www.ebi.ac.uk/ena/browser/view/ERX4964206]

ERX4964203 [https://www.ebi.ac.uk/ena/browser/view/ERX4964203]

SRX7989272 [https://www.ebi.ac.uk/ena/browser/view/SRX7989272]

SRX7989212 [https://www.ebi.ac.uk/ena/browser/view/SRX7989212]

SRX8289612 [https://www.ebi.ac.uk/ena/browser/view/SRX8289612]

SRX1490814 [https://www.ebi.ac.uk/ena/browser/view/SRX1490814]

ERX1301162 [https://www.ebi.ac.uk/ena/browser/view/ERX1301162]

SRX1490815 [https://www.ebi.ac.uk/ena/browser/view/SRX1490815]

SRX7989198 [https://www.ebi.ac.uk/ena/browser/view/SRX7989198] Source data are provided with this paper.

## Code availability

Code and necessary files for recreating downstream analysis, including the intermediate variant mutation table files generated by the snakemake pipeline, can be found at the project github repository (stable release: https://doi.org/10.5281/zenodo.18468386; github: https://github.com/fm-key-lab/Zooscreen).

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

## Acknowledgements

FMK is supported by the Klaus Tschira Foundation (GSO/KT030) and the EASI-Genomics TNA project PID10336. The EASI Genomics project received funding from the European Union's Horizon 2020 research and innovation programme under grant agreement No 824110. FMK, AKWR, and ILM are supported by the Max Planck Society. The taphonomic and palaeopathological work, was carried out by KP as a part of two research internships at the Max Planck Institute (2022 and 2023: ID-UB Mobility programme, Call No. 72 Task 7 to KP). ME was supported by the Praemium Academiae Award of the Czech Academy of Sciences. RK was supported by the Czech Academy of Sciences (RVO:67985912). Excavations at Tilla Bulak were funded by a grant from the Gerda Henkel Foundation, Düsseldorf, Germany (Az. 16/ZA/07 awarded to KK). We thank Rainer Linke (Königsbrunn), Sebastian Gairhos (Stadtarchäologie Augsburg), Marcin Woźniak, Magdalena Miciak, and Monika Kwiatkowska (Poland) for providing access to archaeological specimen. We would like to thank the German Archaeological Institute for providing work space for KP and AKWR. We would also like to thank Meike Sorensen for her help with testing the SCR protocol before it was implemented in the aDNA lab, and for all her support in setting up the lab. We are grateful to Diane Schad for her help with refining the figures. We thank the Key Lab for helpful discussion. Finally, we thank all the people who have been involved in the archaeological excavations from which we obtained samples, as well as those responsible for the curation of each site.

## Author contributions

FMK conceived the project with K.P. providing conceptual input. K.M., S.T., J.E., M.E., R.K., M.H., D.P., A.C., N.B., D.D., J.K., A.N., A.A.K., A.R.K., V.E.M., A.B.B., M.T., S.H., P.W.S., K.K., R.U., S.R., R.E.G., and K.P. curated and analysed the zooarchaeological materials and provided contextual information. S.T., M.H., N.B., K.P., and R.E.G. performed initial zooarchaeological analyses. K.P. performed palaeopathological and taphonomic analyses on the samples from Germany, Poland, Romania and Uzbekistan. E.A.N. performed palaeopathological assessments of the German collection and designed sampling strategies for ancient DNA analysis. C.L.K. and K.B. shared expertise in palaeopathology. A.K.W.R., M.K., H.K., and C.L.S. performed the ancient DNA laboratory work. ILM analysed the data with input from F.M.K. A.K.W.R., I.L.M., K.P., and F.M.K. interpreted the DNA and palaeopathological results. All authors revised and accepted the manuscript prior to publication.

## Funding

## Competing interests

The authors declare no competing interests.

## Additional information

¹Evolutionary Pathogenomics, Max Planck Institute for Infection Biology, Berlin, Germany. ²Charité—Universitätsmedizin Berlin, Berlin, Germany. ³Institute for Pre- and Protohistoric Archaeology and Archaeology of the Roman Provinces, Ludwig Maximilian University Munich, Munich, Germany. ⁴Estonian Biocentre, Institute of Genomics, University of Tartu, Tartu, Estonia. ⁵Department of Environmental Sciences, University of Basel, Basel, Switzerland. ⁶State Office for

Cultural Heritage Management Baden-Württemberg, Osteology Working Group, Konstanz, Germany. [7]Institute of Palaeoanatomy, Domestication Research and the History of Veterinary Medicine, Ludwig Maximilian University Munich, Munich, Germany. [8]Department of Archaeology, Simon Fraser University, Burnaby, BC, Canada. [9]Department for Archaeogenetics, Max Planck Institute for Evolutionary Anthropology, Leipzig, Germany. [10]Department of Anthropology, University of Western Ontario, London, ON, Canada. [11]Curt-Engelhorn-Centre Archaeometry, Mannheim, Germany. [12]Department of Prehistoric Archaeology, Institute of Archaeology, Czech Academy of Sciences, Prague, Czech Republic. [13]Department of Natural Sciences and Archaeometry, Institute of Archaeology, Czech Academy of Sciences, Prague, Czech Republic. [14]Natural Sciences Department, Archaeozoology Laboratory, German Archaeological Institute, Berlin, Germany. [15]Department of Biostructure and Animal Physiology, Faculty of Veterinary Medicine, Wrocław University of Environmental and Life Sciences, Wrocław, Poland. [16]Eurasia-Department, German Archaeological Institute, Berlin, Germany. [17]Central Bohemian Museum in Roztoky, Roztoky, Czech Republic. [18]Independent researcher, Stawropol, Russian Federation. [19]Department of Archaeology, Faculty of History, Lomonosov Moscow State University, Moscow, Russian Federation. [20]Institute of Archaeology RAS, Moscow, Russian Federation. [21]Nasledie Cultural Heritage Unit, Stawropol, Russian Federation. [22]Department of Zoology, University of Cambridge, Cambridge, UK. [23]Vasile Pârvan Institute of Archaeology, Romanian Academy, Bucharest, Romania. [24]Institute of Near Eastern Archaeology, Ludwig-Maximilians-University, Munich, Germany. [25]Department of Anthropology, Dedman College of Humanities and Sciences, Southern Methodist University, Dallas, TX, USA. [26]Department of Palaeoenvironmental Research, Institute of Geology, Adam Mickiewicz University, Poznań, Poznań, Poland. ✉e-mail: ak@palaeome.org; koka@amu.edu.pl; key@mpiib-berlin.mpg.de

