## [Peer Review file · Nature Communications]

Probing the zooarchaeological record across time and space for ancient pathogen DNA

Corresponding Author: Dr Felix Key

Version 0:

Reviewer comments:

Reviewer #1

(Remarks to the Author)

General comments:

This is a very interesting and innovative contribution to the consideration of past animal reservoirs of infectious disease and the articulation of palaeopathological and genetic data. The work will be of interest to a broad range of researchers.

Palaeopathological data presentation:

In the abstract, the manuscript proposes that “Our work presents a pathway to understanding prehistoric zoonotic diseases by integrating zooarchaeological, palaeopathological, and genetic data.” In principle, this is true, but in practice it does not effectively link or present the palaeopathological data. This data can be linked through the unique IDs for individual specimens in the supplementary tables (notably Table S2), but as it stands it is difficult to understand the nature of each pathology in a consistent fashion (see below).

Data on the palaeopathological assemblage seems to be inconsistently presented across the files. For example, for Bürgermeister-Ulrich-Straße 100 in the supplementary information it is stated that: “From the small faunal assemblage, two specimens have been selected for ancient DNA sampling: This includes a thoracic vertebra of a small domestic ruminant showing exostoses at the caudal surface of the Processus thoracalis and a cattle metacarpal with a possible exostosis on the dorsal side of the proximal diaphysis, even if this feature is not clearly visible due to strong bone surface erosion.” (SI, p.11). Whereas in Table S2 the specimens listed as being sampled are noted as having ‘No’ palaeopathology, and the taxonomic host classification data seems slightly at odds with the description in the text. All data across the supporting information text and tables should be checked for consistency. In addition, palaeopathology is sometimes inadequately and inconsistently described. For example, the 15 samples from Giecz 10 are just described as having ‘pathology’ in Table S2, and they are not individually described in the Supplementary Information text. Reviewing the data classification in the ‘Palaeopathology’ column of Table S2 reveals some slightly more detailed descriptions, some that assign a general type of pathology (e.g. arthropathy), and some that simply fall into a yes/no classification. Ideally, pathological data description should be 1) consistent, and 2) based upon precise description (e.g. location, extent and description of changes associated with the lesion) to be useful to other palaeopathology researchers. The authors present a pioneering dataset here, and without these data it is a rather asymmetrical study that falls short of its potential. Given that the sampling is also destructive, there are important ethical reasons why adequate description of lesions should also be included.

A One Health approach:

On page 3 (lines 84-90), the manuscript aligns itself with a One Health approach. This study tends to only emphasize the zoonotic risks posed by animals, rather than also considering the potential for reverse zoonoses (humans-animals) and infections between different animals species also (all key dynamics in multispecies farming systems). There is a wider consideration of One Health in the archaeological literature that might be worth considering here (e.g. Rayfield et al. 2023, <https://doi.org/10.1098/rspb.2023.0525>) to strengthen the theoretical context for this work.

Palaeopathology methods:

The statement on palaeopathological methodology is too brief and should be more fully developed: “The analysis of pathology was conducted using a wide range of references and classifications 19,58 lesions that displayed signs of an active infection 19,58” (page 17, lines 480-481). References 19 (Baker and Brothwell 1980) and 58 (Bartosiewicz and Gal 2013) are useful general textbooks on palaeopathology, but this is not a full and effective statement on how pathology was recorded and interpreted.

Additional comments:

Page 5, lines 139-142

This sentence sounds a little discordant, with Neolithization described as occurring in the Bronze Age.

Page 11, lines 298-299

“However, whether the palaeopathological lesions and the ancient bacterial DNA signatures can be directly linked is unknown.”. Why is this? Is it that you mean that the precise pathological expression recorded cannot be linked to a specific bacterial disease (because such infections tend to be non-specific), rather than meaning that the data can't be linked. As it stands it is a little unclear - this statement should be clarified and explained more fully.

Page 11, line 304

“No other site was enriched for samples with pathogen signatures.”. Meaning unclear – rephrase?

Page 11, lines 316-7

“In sum, despite being mostly unspecific about the precise bacterial species, palaeopathological lesions can guide the prioritisation of promising bone specimens for the genomic investigation of pathogens.”. I agree, and it is partly for this reason that the pathological descriptions supporting this piece of work should be improved.

Page 16, lines 427-428

“Bone remodelling is generally a slow process; although, some acute infections can cause skeletal changes, i.e., acute osteomyelitis.” Is it the case that you are suggesting the more rapid changes associated with some acute infections – if so, it might be useful to make this point more clearly.

Page 17, lines 486-488

“Their different chronologies made it possible to capture changes over time and, in particular, the timing of pathogen transmission in the human–animal relationship.” You are only sampling animals here, so not actually capturing the transmission of pathogens between animals and humans (but the potential for this). Perhaps rephrase.

Page 17, lines 488-489

“The geographic range of the sites could help indicate the potential location of the emergence of zoonotic disease.” I'm not sure the coverage is sufficient to identify location of the emergence of specific diseases. Perhaps rephrase.

(Remarks on code availability)

Reviewer #2

(Remarks to the Author)

The authors perform palaeopathological and metagenomic analyses on 346 skeletal elements from both domesticated and wild animals collected from 34 Eurasian archaeological sites dating from 4650/4350 BCE to 900-1200 CE, with the aim of assessing the detectability and DNA recovery potential of ancient pathogens in zooarchaeological assemblages. The study presents a valuable and comprehensive dataset, incorporating a wide range of species, skeletal elements, and pathological lesions in its analysis. Whilst the chronological distribution is heavily centred on the Bronze Age (65% of the sites) and the assemblage studied primarily consists of bones displaying lesions (88% of the total bones), it fills a significant gap in the literature as a majority of ancient pathogen studies are focused on human rather than animal remains. This makes it a valuable and timely contribution to the fields of zooarchaeology, palaeogenomics, metagenomics, and historical epidemiology, offering important insights for future research.

However, while the study's breadth is commendable and both the palaeopathological and metagenomic analyses are methodologically sound and appropriate, the overarching aim of the research remains somewhat unclear. This lack of clarity makes it difficult to fully evaluate the appropriateness of the methodology (primarily in terms of chosen dataset and statistics) and significance of some of the statements in relation to the intended objectives. A more explicit articulation of the primary research questions, a clearer explanation of how the methodology addresses those questions, and a more structured presentation of the discussion around these aims would substantially improve the manuscript's coherence and overall impact.

I recommend publication pending revisions focused on strengthening the framing of the study, and providing additional clarification of several key statements.

Main comments:

1) The primary aim of the paper is not immediately clear:

The authors provide a comprehensive introduction on the current state of ancient pathogen research based on the zooarchaeological record: the increase in zoonoses during the Neolithic, the relatively recent application of ancient pathogen analysis to zooarchaeological assemblages, and the challenges of such investigations linked to the nature of human-animal interactions in the past when compared to similar research conducted on ancient human populations only.

In the last paragraph of the introduction, line 122, the authors state: "In this study, we investigate hundreds of zooarchaeological specimens from across Eurasia for pathogen DNA, with a particular focus on the Bronze Age. We use palaeopathology to identify lesions that may have resulted from infectious diseases. We use these lesions as targets for ancient DNA sampling to increase the probability of recovering ancient pathogen DNA and identify known pathogens that were present in prehistoric animal populations."

In the first paragraph of the results section, the authors then mention "To test the preservation of microbial pathogen DNA within the record" and "The sites cover periods dating to the Neolithic to the Medieval period, with the majority of sites belonging to the Bronze Age. We focused on investigating Bronze Age zooarchaeological specimens because human-derived ancient zoonotic-pathogen genomes were repeatedly identified in specimens from the Bronze Age, a period of major human migratory events and the Neolithization in Eurasia"

And in the second paragraph of the discussion, it is stated that "Our results suggest that leveraging samples with palaeopathological lesions for genomic analysis provides a focused and valuable method for identifying ancient pathogens from zooarchaeological remains."

Based on the comprehensive review provided in the introduction, and the study design mentioned in these paragraphs, the specific aim of the study remains ambiguous. Is the study primarily aimed at: * Assessing the potential for recovering pathogen DNA from faunal remains that exhibit pathological lesions?

* Testing the hypothesis that zoonotic disease frequency increased during the Neolithic and Bronze Age?

* Exploring the overall preservation and detectability of microbial pathogen genomes in zooarchaeological specimens? In which case, exploring a specific time period or comparing between time periods?

Each of these objectives implies a distinct methodological approach. In addition, although details are provided in the results, precision on the number of samples investigated needs to be given in the introduction: the term "hundreds" is vague, potentially ranging anywhere from 300 to 900 or more.

2) Several interesting observations are made throughout the paper, but they appear restricted to their own sections, and are not further developed or integrated into the broader discussion. Depending on the overarching aim of the paper and its sub-aims/objectives, they should be addressed in the results and further discussed in the Discussion section. For instance:

2a) Lines 289-290, the authors mention: "We recovered DNA from *P. gingivalis* from a tooth, underlying the history of periodontal disease among human-associated livestock."

This is an exciting find, yet apart from this sentence, no accompanying discussion is given, not even the genome coverage, or the host or the archaeological site from which that tooth originates (information also not available in the supplementary tables). This rarely qualifies as "underlying the history of periodontal disease among human-associated livestock." It would warrant at least more details about the identification of this pathogen within the analysis, and offer some insights into future research.

2b) Line 474: "Although human (brachydont) teeth have been shown to preserve DNA from blood-borne pathogens, it is unclear if the same is true in species that do not have enclosed pulp chambers, such as ruminants (hypsodonts)." This is a very interesting point and one that would be worth exploring given the dataset here used, but not further reference to it or discussion is made anywhere in the paper.

2c) The identification of *S. enterica* within the studied assemblage is briefly explored as a case study in the Discussion section, but the results are not presented or addressed in the Results section, apart from lines 283-285: "The pathological changes in the mandibles that produced bacterial hits included periodontitis, abscesses, and antemortem tooth loss, with genomic signatures for *Bordetella*, *Corynebacterium*, *Enterococcus*, *Salmonella*, *Staphylococcus*, and *Streptococcus*.". The presence of *S. enterica* should be more explicitly reported and contextualized within the Results section, especially given the importance of this pathogen today.

3) I understand that the study targets species pathogenic to humans and animals, including opportunist pathogens able to colonise hosts without causing disease. As stated in the Materials and Methods, lines 546-550: "A list of taxa of candidate pathogen species and genera can be found in (Table S3), which includes known zoonotic pathogens and genera, along with additional animal-specific pathogens. We removed the genus node of *Bacillus*, *Clostridium* and *Brucella* due to closely related non-pathogenic species highly abundant in soil that complicates the identification of positive samples."

However, I feel further clarification is needed regarding the selection and curation of the reference database used for

bacterial, viral, and parasitic taxa.

As the authors used the HOPS pipeline, was the HOPS default pathogen list (https://github.com/rhuebler/HOPS/blob/external/Resources/default_list.txt), which comprised 356 entries, also initially used and further treated and/or complemented with other species of interest? Table S3 lists 266 entries. Note that the HOPS default pathogen list includes some duplicate entries, taxa classified only at the family or genus level without a corresponding species-level reference genome, as well as certain sub-strains and incomplete genomes. Were such entries filtered out or otherwise treated differently during the analysis?

Moreover, while Table S3 lists the species of interest to the study, it does not specify the exact reference genomes used or their accession numbers. This information needs to be added for reproducibility purposes.

4) In the section on 'Interpretation of DNA evidence in its palaeopathological context', the authors point out that 23.3% of all bones with lesions produced robust bacterial hits. They then mention that the "fact that no ancient pathogenic bacteria were authenticated from the 25 bones sampled without any detectable lesions (p -value <0.005 , Chi square test) emphasizes the advantage of palaeopathological investigations for prioritisation of specimens."

Firstly, I think the difference in sample size (189 bones with visible palaeopathological lesions versus 25 bones with no visible palaeopathological lesions) needs to be clearly acknowledged. If the primary aim of the paper was to test and evaluate the advantage of using skeletal elements with pathological lesions compared to samples without, a more balanced dataset would have been needed. The authors recognise this lines 424-426: "Future investigations explicitly designed with comparable amounts of pathological and non-pathological specimens per site will be instrumental to further corroborate our findings." But it needs to be explicitly stated beforehand.

Secondly, while the reported Chi-square test yields a significant result ($p < 0.005$) for the samples without any detectable lesions, the small sample size and the fact that no ancient pathogenic bacteria was authenticated ($n=0$) suggest that a Fisher's exact test may be more appropriate in this context. I would recommend reporting results from a Fisher's test as a robustness check, or at least discussing the statistical limitations of using Chi-square under these conditions, especially if this is part of the main findings of the paper, which it appears to be based on the second paragraph of the discussion (lines 397-399; lines 412-414).

5) Lines 486-487: " Their different chronologies made it possible to capture changes over time and, in particular, the timing of pathogen transmission in the human–animal relationship". Similarly to point 4, this is misleading considering the majority of the samples come from the Bronze Age (65%) and no statistical tests were undertaken between the different time periods to support this statement. This also leads to confusion in relation to the aim of the paper, which according to this statement, was to evaluate if increases or decreases of pathogens could be observed over time (this is also further complicated by the fact that there are no reports on all the pathogens identified for each site, if this was indeed the aim of the paper).

6) The term "One Health" is listed as the first keyword, and is briefly explained in the opening paragraph of the introduction. However, this research does not engage with the One Health framework. While it includes both domestic and wild animals and targets zoonotic pathogens, the analysis and discussion are centred exclusively on domesticated species, with no reference to human or environmental health. As such, I recommend removing "One Health" as a keyword as it is misleading, unless the authors expand the discussion to more clearly align the study with this interdisciplinary framework.

7) The first two lines of the Materials and Methods mention: "The study materials were animal bones and teeth. An initial archaeozoological assessment had been carried out for all the faunal assemblages." This is very limited information for a Materials and Methods section. The study material needs to be described in greater detail here, including the number of bones and teeth analysed, and the time periods covered - similar to what is outlined in the Results section.

Furthermore, as the zooarchaeological analysis represents the baseline of this study - providing species identification through morphology and recording bone surface modifications (e.g. taphonomic marks, cut marks, and of course pathological lesions), a more comprehensive description of the zooarchaeological methodology is essential. I acknowledge that there are references to some manuals, but given how key the zooarchaeological analysis is and for purposes of clarity and reproducibility, these should be clearly detailed.

Minor comments :

8) Lines 214-216, the authors state: "From the 346 skeletal elements, we produced a total of 357 DNA extracts. These included 20 subsamples collected from 11 bones as well as nine teeth collected from four individuals".

Having looked through Table S2, I have identified 11 teeth from four individuals:

KOK-009 (3 teeth)

AUTAWE111-004 (4 teeth)

AUTAWE85-003 (2 teeth)

AUTAWE85-023 (2 teeth)

9) Lines 216-217: "Samples were sequenced on the Illumina platform, which, apart from five failed samples, generated between 2,324,302 and 90,823,912 sequencing reads per sample (Table S2)."

Only 4 samples have N/A, unless sample ID 353/17.F1 is also a fail, in which case I would clarify it in the notes column.

10) Line 304: "No other site was enriched for samples with pathogen signatures."

The term 'enriched' within a metagenomic and palaeogenomic study is misleading. I would use another term.

11) Lines 338-341: "Ascertaining 11,970 mutations among the outgroup *E. tonsillarum*, a previously human-derived medieval *E. rhusiopathiae* genome and 42 *E. rhusiopathiae* genomes isolated from a diverse set of contemporary hosts, [...]" There are however 3 previously human-derived medieval *E. rhusiopathiae* genomes on the phylogenetic tree (Figure 5), and a total of 7 in Table S4.

Furthermore, in the caption of Figure 5: "Among 42 *E. rhusiopathiae* genomes and a single *E. tonsillarum* (outgroup) genome we ascertained 11,970 mutations. For four ancient genomes (AZP-012 and three previously published human-derived genomes) the nucleotide call was inferred for each of the mutations."

It is here confusing which dataset was used for the ascertainment process. And if performed on only a single human-derived sample, as mentioned in the text, could the author provide the corresponding sample ID.

12) Lines 360-361: "In order to place the ancient genomic information into the modern genetic diversity, we leveraged genomes of *S. lutetiensis*, *S. infantarius*, and *S. gallolyticus* isolated from different sources across Eurasia (Table S4)." To enhance clarity, could the authors state the total number of genomes used for the ascertainment process, even if this information is already included in the figure caption and Table S1?

Table S4 lists 21 genomes in addition to the three ancient samples from this study. These include *S. lutetiensis*, *S. gallolyticus*, and *S. infantarius*, but also include the outgroup *S. salivarius* and five *S. equinus* genomes. This differs from what is mentioned in the text. Could the author clarify this?

13) Lines 394-396: "In this study, we applied a targeted approach to explore the suitability of zooarchaeological remains that primarily displayed palaeopathological lesions, which could have resulted from infection, for genomic palaeomicrobiological analysis"

In the context of metagenomic studies, the term 'targeted approach' can be misinterpreted as referring to the molecular targeting strategy (e.g. enrichment and capture), rather than the intended meaning of targeting particular skeletal elements or specimens. I recommend rephrasing this sentence to avoid ambiguity.

Typos/Wording :

Line 58: Should it not be 'Neolithic' rather than 'prehistory'?

Line 138: "The sites cover periods dating from the Neolithic to the Medieval period [...]"

Line 140/141: "[...]because human-derived ancient zoonotic pathogen genomes have been repeatedly identified in the literature in specimens from the Bronze Age."

Line 305: "While we observe...[...]" remove comma

Line 316: "In summary, despite..."

Line 427: "Bone remodelling is generally a slow process 50, although some acute infections [...]" remove semi-colon, add comma

Line 436-437: Dot is at the beginning of line 437, needs to be at end of 436.

(Remarks on code availability)

Reviewer #3

(Remarks to the Author)

This is an excellent piece of work by Felix Key's team. A large part of this paper's novelty lies in showcasing querying zooarchaeological assemblages for pathogen DNA, and also this paper offers a fresh perspective on focusing on pathogens that are not exclusively human, nor centered around historic human epidemics. They have employed a robust set of established methods, as well as developed custom pipelines, to metagenomically identify pathogens in ancient DNA data and perform robust phylogenetic placements of low coverage samples in modern phylogenetic trees. I highly recommend this paper gets published.

Here is a list of minor comments that could help polish this manuscript even more:

Intro

Lines 71-79: Even though I understand what the authors are trying to communicate this piece of text feels a bit choppy and "rushed" in terms of introducing all these ideas. Maybe moving the sentence starting with "For instance, the 1918 flu pandemic ..." right before the previous sentence ("Such zoonotic spillover events can ...") would improve the fluidity of that section.

Results

Lines 150-151: The authors mention in that figure legend that "Note that the Bronze Age covers different absolute chronological horizons within different geographical locations." This is a really important point and it would be worth including in the main text, as this might not be an apparent piece of info for readers not accustomed to ancient DNA and/or scientific archaeology.

Lines 279-281: This is a cool and reasonable result. Could the authors maybe provide an explanation/speculation for the reason in the discussion? e.g. could it be because these bones are denser than others, or could it be because these are the more frequently preserved skeletal elements when buried?

Lines 294-296: This is a rather interesting but unexpected result. It is true that although *E. coli* and *Enterococcus* are mainly gut commensals, they can be found in other body parts or even free-living in the environment. Finding them in ribs would imply (at least naively) either acute bacteremia or a lung infection (which is rather unusual). Could the authors elaborate a tad more on that result and its interpretation, since they explained very well in their discussion the difficulty of distinguishing a true ancient pathogen from "ancient-like" contaminant species that might have colonised the bone after the host's death?

Lines 313-315: This is a really interesting result. Could the authors elaborate on it in the discussion and provide a hypothesis of why that happened?

Figures 5 and 6: I am curious why the authors chose to include modern genomes that did not have full metadata on the host and the country of origin? Was it because there were not enough high quality modern genomes with their metadata filled in that fit their criteria? If so it could be really helpful to make a note of it.

Methods:

The "Pathogen phylogenetic reconstruction", although detailed could be written in a simpler/clearer way so that it is easier to follow through the individual steps the authors took.

(Remarks on code availability)

The GitHub repo linked to this manuscript contains all the necessary code and scripts to reproduce the analyses presented in this paper

Version 1:

Reviewer comments:

Reviewer #1

(Remarks to the Author)

The authors have responded appropriately and precisely to my previous comments and I am happy to support the manuscript being published. It is an excellent piece of work, which will be of interest to a broad range of researchers.

(Remarks on code availability)

Reviewer #2

(Remarks to the Author)

The revisions made to the manuscript have greatly improved both its clarity and coherence, with the overall aim of the study now explicitly stated. By combining palaeopathological and genomic analyses across a comprehensive dataset of 346 skeletal elements from both domesticated and wild animals, recovered from 34 Eurasian sites spanning the last 7,000 years and a wide range of environments, the authors provide well-supported recommendations for sampling strategies for ancient pathogen DNA research within the zooarchaeological record. Archaeological faunal specimens remain a resource vastly understudied within the field of palaeomicrobiology, and their overall potential has remained unaddressed until now. As such, this study represents an essential guide for future palaeopathological and epidemiological investigations based on ancient faunal remains.

The authors identified several bacteria taxa, including *S. enterica*, *C. burnetii*, and *P. gingivalis*, and present two case studies focusing on *E. rhusiopathiae* and *S. lutetiensis*, in which both the ancient nature of the genomic data is authenticated, and the evolutionary relationships between these selected low-coverage ancient genomes and their present-day relatives are examined. Overall, all major findings are addressed in sufficient detail, and limitations are acknowledged where appropriate.

The results and discussion are well aligned with the study's original aim. The discrepancies between the text and the tables have been addressed, and the manuscript reads with a clear and cohesive narrative. I thank the authors for such careful and thorough revision. I would also like to commend the high level of transparency, particularly regarding data and code availability, which is essential for ensuring the reproducibility and reliability of the work.

Overall, all comments have been satisfactorily addressed, and I believe the manuscript represents a strong and valuable contribution to the field and is suitable for publication. I have no major comments, only three minor points intended more for clarification.

1. In the abstract, it is mentioned that "68 signatures of ancient (opportunistic) pathogens" have been identified (line 69). Could the authors clarify what this refers to? According to Figure 4, I understand that 29 bacterial pathogens have been

identified across the 55 libraries that produced high-confidence bacterial hits, of which 24 libraries displayed hits from multiple bacteria.

2. Lines 260-261, lines 305-308 and lines 676-677, the authors refer to Figure 3 when discussing the identification of signatures from various bacteria including *E. rhusiopathiae*, *S. lutetiensis*, *Bordetella*, *Corynebacterium*, *Enterococcus*, *Salmonella*, *Staphylococcus*, and *Streptococcus*. Yet Figure 3 illustrates examples of pathological lesions sampled in this study and which produced pathogen signatures for the screening, but it does not explicitly indicate the bacterial taxa. I was wondering if the authors intended to refer to Figure 4, which clearly presents the full range of identified bacterial taxa?

3. In Table S4, seven ancient *E. rhusiopathiae* Iberian genomes are listed. However, in the main text (line 699), it is mentioned that eight additional publicly available ancient metagenomes were reanalysed. I think there may be a discrepancy?

(Remarks on code availability)

Reviewer #1 (Remarks to the Author):

General comments:

This is a very interesting and innovative contribution to the consideration of past animal reservoirs of infectious disease and the articulation of palaeopathological and genetic data. The work will be of interest to a broad range of researchers.

Palaeopathological data presentation:

1. In the abstract, the manuscript proposes that “Our work presents a pathway to understanding prehistoric zoonotic diseases by integrating zooarchaeological, palaeopathological, and genetic data.” In principle, this is true, but in practice it does not effectively link or present the palaeopathological data. This data can be linked through the unique IDs for individual specimens in the supplementary tables (notably Table S2), but as it stands it is difficult to understand the nature of each pathology in a consistent fashion (see below).

2. Data on the palaeopathological assemblage seems to be inconsistently presented across the files. For example, for Bürgermeister-Ulrich-Straße 100 in the supplementary information it is stated that: “From the small faunal assemblage, two specimens have been selected for ancient DNA sampling: This includes a thoracic vertebra of a small domestic ruminant showing exostoses at the caudal surface of the Processus thoracalis and a cattle metacarpal with a possible exostosis on the dorsal side of the proximal diaphysis, even if this feature is not clearly visible due to strong bone surface erosion.” (SI, p.11). Whereas in Table S2 the specimens listed as being sampled are noted as having ‘No’ palaeopathology, and the taxonomic host classification data seems slightly at odds with the description in the text. All data across the supporting information text and tables should be checked for consistency.

3. In addition, palaeopathology is sometimes inadequately and inconsistently described. For example, the 15 samples from Giecz 10 are just described as having ‘pathology’ in Table S2, and they are not individually described in the Supplementary Information text. Reviewing the data classification in the ‘Palaeopathology’ column of Table S2 reveals some slightly more detailed descriptions, some that assign a general type of pathology (e.g. arthropathy), and some that simply fall into a yes/no classification. Ideally, pathological data description should be 1) consistent, and 2) based upon precise description (e.g. location, extent and description of changes associated with the lesion) to be useful to other palaeopathology researchers. The authors present a pioneering dataset here, and without these data it is a rather asymmetrical study that falls short of its potential. Given that the sampling is also destructive, there are important ethical reasons why adequate description of lesions should also be included.

Thanks for catching inconsistencies among our paleopathological data presentation. We agree that palaeopathology was sometimes inadequately and inconsistently described. We have carefully checked

the supplementary information for each site to make sure it aligns with the updated Table S2. Moreover, we substantially expanded the paleopathological information, which now includes location (skeletal element), extent, and a description (of changes associated with the lesion) consistently presented in Table S2 for every specimen with identified paleopathological lesions.

4.A One Health approach:

On page 3 (lines 84-90), the manuscript aligns itself with a One Health approach. This study tends to only emphasize the zoonotic risks posed by animals, rather than also considering the potential for reverse zoonoses (humans-animals) and infections between different animals species also (all key dynamics in multispecies farming systems). There is a wider consideration of One Health in the archaeological literature that might be worth considering here (e.g. Rayfield et al. 2023, <https://doi.org/10.1098/rspb.2023.0525>) to strengthen the theoretical context for this work.

Following the reviewers comments we have further elaborated the manuscript placement within the wider field of the One Health concept. The goal of our work is to show that pushing paleomicrobiology into the zooarcheological record will help to expand the One Health concept into pre-history. We thank the reviewer for pointing out the review by Rayfield 2023, which discusses the approach, and which we now added to the manuscript. We have adjusted the manuscript to reflect this goal of our manuscript broadly, incl. its potential contribution to identifying reverse zoonoses and animal-animal transmission. See also comment #25.

Introduction line 86: “In addition, the intensification of inter-species interactions between humans and their domesticated animals during livestock management, butchering, or consumption of animal-derived products increased the likelihood of zoonotic, reverse zoonotic, and animal to animal transmission events (Bartosiewicz 2022; Rayfield et al. 2023).”

Introduction line 96: “While screening for microbes in large genomic datasets generated for ancient human population studies has become reasonably routine, the same cannot be said for ancient faunal datasets. As a result, limited information is available concerning disease reservoirs in ancient and historic animal populations - posing a barrier to expand the One Health concept into the past.”

Discussion Line 513: “Hence, this study is an important step towards exploring ancient pathogens in the zooarchaeological record, necessary to expand the One Health concept into the past, which holds the promise to elucidate the reservoir and host range of zoonotic pathogens, the geographic and temporal spread, and the genetic mechanisms enabling the evolutionary adaptation towards the human host..”

5.Palaeopathology methods:

The statement on palaeopathological methodology is too brief and should be more fully developed: “The analysis of pathology was conducted using a wide range of references and classifications 19,58 lesions that displayed signs of an active infection 19,58” (page 17, lines 480-481). References 19 (Baker and

Brothwell 1980) and 58 (Bartosiewicz and Gal 2013) are useful general textbooks on palaeopathology, but this is not a full and effective statement on how pathology was recorded and interpreted.

We agree with the reviewer. The palaeopathological methodology has been expanded to provide a more comprehensive overview. See page 17 lines 520-550.

Additional comments:

6. Page 5, lines 139-142

This sentence sounds a little discordant, with Neolithization described as occurring in the Bronze Age.

We have changed the sentence to clarify that the Bronze Age followed the Neolithic.

7. Page 11, lines 298-299

“However, whether the palaeopathological lesions and the ancient bacterial DNA signatures can be directly linked is unknown.”. Why is this? Is it that you mean that the precise pathological expression recorded cannot be linked to a specific bacterial disease (because such infections tend to be non-specific), rather than meaning that the data can’t be linked. As it stands it is a little unclear - this statement should be clarified and explained more fully.

The sentence was meant to convey that although we recovered the bacterium from the lesion, we cannot say that this bacterium caused the infection that gave rise to the lesion. We have deleted this sentence from this section as it is better covered in the discussion.

Line 458: “It is also worth noting that identified bacterial hits often cannot be conclusively linked to the observed lesions, which tend to be non-specific. The bacteria may also represent opportunistic colonisation of an existing site of infection or they could have been transferred to the sample in the depositional environment from another infected source.”

8. Page 11, line 304

“No other site was enriched for samples with pathogen signatures.”. Meaning unclear – rephrase?

We improved the section's clarity.

Line 329: “Although the specimens collected from this site constitute 29% of the entire dataset, the number of positive bacterial hits (58.2% of total samples with bacterial hits) is disproportionate compared to other sites (p-value<0.00002; Fisher Exact test). No other sites had disproportionate pathogen recovery that rose to statistical significance.”

9. Page 11, lines 316-7

“In sum, despite being mostly unspecific about the precise bacterial species, palaeopathological lesions can guide the prioritisation of promising bone specimens for the genomic investigation of pathogens.”. I

agree, and it is partly for this reason that the pathological descriptions supporting this piece of work should be improved.

Thank you for this comment. We hope the improvements to the pathological descriptions are acceptable (see response to comment 1-3 and 5).

10. Page 16, lines 427-428

“Bone remodelling is generally a slow process; although, some acute infections can cause skeletal changes, i.e., acute osteomyelitis.” Is it the case that you are suggesting the more rapid changes associated with some acute infections – if so, it might be useful to make this point more clearly.

We have specified that acute osteomyelitis is a bone infection that develops quickly (within weeks). We mention it to highlight that there are exceptions to the generally slow process of bone remodelling.

Line 455: “The majority of palaeopathological lesions are the result of long term and chronic infections as bone remodelling is generally a slow process. Although some infections, like acute osteomyelitis, can rapidly induce skeletal changes, individuals who perished from acute infections are often not visibly identifiable in the archaeological record.”

11. Page 17, lines 486-488

“Their different chronologies made it possible to capture changes over time and, in particular, the timing of pathogen transmission in the human–animal relationship.” You are only sampling animals here, so not actually capturing the transmission of pathogens between animals and humans (but the potential for this). Perhaps rephrase.

The sentence has been rephrased in accordance with our study.

Line 526: “Second, their different chronologies may capture pathogen emergence within animal populations over time.”

12. Page 17, lines 488-489

“The geographic range of the sites could help indicate the potential location of the emergence of zoonotic disease.” I’m not sure the coverage is sufficient to identify location of the emergence of specific diseases. Perhaps rephrase.

The sentence has been rephrased in accordance with our study.

Line 527: “Third, the geographic range of sites with samples positive for ancient pathogen DNA can help indicate the range of past zoonotic disease.”

Reviewer #2 (Remarks to the Author):

The authors perform palaeopathological and metagenomic analyses on 346 skeletal elements from both domesticated and wild animals collected from 34 Eurasian archaeological sites dating from 4650/4350 BCE to 900-1200 CE, with the aim of assessing the detectability and DNA recovery potential of ancient pathogens in zooarchaeological assemblages. The study presents a valuable and comprehensive dataset, incorporating a wide range of species, skeletal elements, and pathological lesions in its analysis. Whilst the chronological distribution is heavily centred on the Bronze Age (65% of the sites) and the assemblage studied primarily consists of bones displaying lesions (88% of the total bones), it fills a significant gap in the literature as a majority of ancient pathogen studies are focused on human rather than animal remains. This makes it a valuable and timely contribution to the fields of zooarchaeology, palaeogenomics, metagenomics, and historical epidemiology, offering important insights for future research.

13. However, while the study's breadth is commendable and both the palaeopathological and metagenomic analyses are methodologically sound and appropriate, the overarching aim of the research remains somewhat unclear. This lack of clarity makes it difficult to fully evaluate the appropriateness of the methodology (primarily in terms of chosen dataset and statistics) and significance of some of the statements in relation to the intended objectives. A more explicit articulation of the primary research questions, a clearer explanation of how the methodology addresses those questions, and a more structured presentation of the discussion around these aims would substantially improve the manuscript's coherence and overall impact.

I recommend publication pending revisions focused on strengthening the framing of the study, and providing additional clarification of several key statements.

Main comments:

14.1) The primary aim of the paper is not immediately clear:

The authors provide a comprehensive introduction on the current state of ancient pathogen research based on the zooarchaeological record: the increase in zoonoses during the Neolithic, the relatively recent application of ancient pathogen analysis to zooarchaeological assemblages, and the challenges of such investigations linked to the nature of human-animal interactions in the past when compared to similar research conducted on ancient human populations only.

In the last paragraph of the introduction, line 122, the authors state: "In this study, we investigate hundreds of zooarchaeological specimens from across Eurasia for pathogen DNA, with a particular focus on the Bronze Age. We use palaeopathology to identify lesions that may have resulted from infectious diseases. We use these lesions as targets for ancient DNA sampling to increase the probability of recovering ancient pathogen DNA and identify known pathogens that were present in prehistoric animal populations."

In the first paragraph of the results section, the authors then mention “To test the preservation of microbial pathogen DNA within the record” and “The sites cover periods dating to the Neolithic to the Medieval period, with the majority of sites belonging to the Bronze Age. We focused on investigating Bronze Age zooarchaeological specimens because human-derived ancient zoonotic-pathogen genomes were repeatedly identified in specimens from the Bronze Age, a period of major human migratory events and the Neolithization in Eurasia”

And in the second paragraph of the discussion, it is stated that “Our results suggest that leveraging samples with palaeopathological lesions for genomic analysis provides a focused and valuable method for identifying ancient pathogens from zooarchaeological remains.”

Based on the comprehensive review provided in the introduction, and the study design mentioned in these paragraphs, the specific aim of the study remains ambiguous. Is the study primarily aimed at:

- * Assessing the potential for recovering pathogen DNA from faunal remains that exhibit pathological lesions?
- * Testing the hypothesis that zoonotic disease frequency increased during the Neolithic and Bronze Age?
- * Exploring the overall preservation and detectability of microbial pathogen genomes in zooarchaeological specimens? In which case, exploring a specific time period or comparing between time periods?

Each of these objectives implies a distinct methodological approach. In addition, although details are provided in the results, precision on the number of samples investigated needs to be given in the introduction: the term “hundreds” is vague, potentially ranging anywhere from 300 to 900 or more.

We thank the reviewer for the overall positive evaluation and appreciate the comments to carve out more targeted the aims of our study. Our primary objective is to assess the potential for recovering pathogen DNA from faunal remains, focused through the targeted investigation of samples carrying palaeopathological lesions or teeth, as both types showed successful pathogen recovery in ancient humans. Moreover, temporally we focus on prehistoric specimens, in particular the Bronze Age period, because, as known from human ancient palaeomicrobiological investigations, zoonotic diseases seem to become overall more detectable within that period, which means - importantly - the chance to address our primary objective increases. Following the reviewers highlighted sections and throughout the manuscript we have revised the text accordingly.

Last paragraph of the introduction, line 141: “In this study, we investigate whether a selective sampling approach based on palaeopathological analysis and teeth can provide useful targets for ancient pathogen DNA sampling and overcome the challenges in faunal palaeomicrobiology. We investigate 346 zooarchaeological specimens from across Eurasia for pathogen DNA, with a particular focus on the Bronze Age, a promising period for ancient zoonotic pathogen recovery based on human pathogen investigations. Our results show ancient DNA signatures of known (opportunistic) pathogens in domesticated animals, and highlight the power of palaeopathology in prioritising specimens for pathogen DNA recovery, opening a new direction in palaeomicrobiology.”

First paragraph of the results section, line 152: “To understand whether palaeopathological lesions or teeth provide useful targets for recovering ancient microbial pathogen DNA from zooarchaeological remains,

we collected a total of 346 skeletal elements from 329 individual animals discovered at 34 archaeological sites across Eurasia (Figure 1a, Table S1). The majority of these sites are located in Europe, but two sites, Monjukli Depe and Tilla Bulak, are located in Central Asia. The sites span approximately 5,800 years of human history (Figure 1b), with the oldest site, Monjukli Depe, dated to 4650-4350 BCE and the youngest site, Giecz 10, dated to 900-1200 CE. The sites cover periods dating from the Neolithic to the Medieval period, with the majority of sites belonging to the Bronze Age (Table S1). These periods took place at different absolute chronological horizons due to spatiotemporal differences in material culture development within discrete geographical locations. We focused on Bronze Age specimens because human-derived ancient zoonotic pathogen genomes have been repeatedly identified in the literature in specimens from this period, a time of major human migratory events following the Neolithisation in Eurasia.”

First paragraph of the discussion, line 428: “Prehistoric disease reservoirs in faunal populations and the likely increase in pathogen exposure that followed animal domestication are poorly understood and challenging to investigate. In this study, we combined palaeopathological analysis and ancient genomics to explore whether a selective sampling approach focused on palaeopathological lesions indicative of infection or teeth provide a useful starting point for ancient DNA sampling for faunal palaeomicrobiological analysis.”

15. 2) Several interesting observations are made throughout the paper, but they appear restricted to their own sections, and are not further developed or integrated into the broader discussion. Depending on the overarching aim of the paper and its sub-aims/objectives, they should be addressed in the results and further discussed in the Discussion section. For instance:

Following the reviewers comment, we expanded the detail provided in the results as well as the discussion. Generally, and despite the stringent criteria applied, we deliberately do not overemphasize singular signatures identified as more targeted data is required for complete authentication.

16. 2a) Lines 289-290, the authors mention: “We recovered DNA from *P. gingivalis* from a tooth, underlying the history of periodontal disease among human-associated livestock.”

This is an exciting find, yet apart from this sentence, no accompanying discussion is given, not even the genome coverage, or the host or the archaeological site from which that tooth originates (information also not available in the supplementary tables). This rarely qualifies as “underlying the history of periodontal disease among human-associated livestock.” It would warrant at least more details about the identification of this pathogen within the analysis, and offer some insights into future research.

We have added details about the samples from which we obtained the *P. gingivalis* hits to the results and included a discussion of them.

Results, line 310: “We recovered DNA from *P. gingivalis* from a sheep tooth, AZP-123, from Pietrele and a sheep radius, AZP-289, with periostitis from Tilla Bulak.”

Discussion, line 482: “Similarly, we did not obtain sufficient genome coverage to further investigate the two sheep-derived hits to *P. gingivalis*, a commensal oral bacterium that is also a member of the polymicrobial ‘red complex’ associated with periodontal disease in humans and sheep. Although this bacterium is most frequently recovered from the oral cavity, one of our hits originated from a palaeopathological lesion on a femur, which may represent a case of localised bone infection, however contamination cannot be ruled out. Previous research revealed a shift in the composition of the oral microbiota following the Neolithic Transition with agriculturalist populations having a greater number of taxa associated with periodontal disease, including *P. gingivalis*. In the future, animal-derived *P. gingivalis* genomes may help clarify the history of periodontal disease among human-associated livestock as well as potential zoonotic and reverse zoonotic spillovers.”

17. 2b) Line 474: “Although human (brachydont) teeth have been shown to preserve DNA from blood-borne pathogens, it is unclear if the same is true in species that do not have enclosed pulp chambers, such as ruminants (hypsodonts).” This is a very interesting point and one that would be worth exploring given the dataset here used, but not further reference to it or discussion is made anywhere in the paper.

Although this study was not designed to investigate pathogen DNA preservation in different types of teeth, some general observations did indeed emerge from the dataset. We have added this to the results section “Interpretation of DNA evidence in its palaeopathological context” as well as the discussion.

Results, line 312: “While blood-borne pathogens are frequently recovered from human (brachydont) teeth 33, it is unclear if the same is true for species that do not have enclosed pulp chambers, such as ruminants (hypsodonts 45). Of the 11 teeth that produced bacterial hits, four belonged to dogs (brachydont; 12.5% of total dog teeth), while 7 belonged to hypsodonts (cattle, sheep and goats; 8% of total teeth from those species), showing no significant difference in the identification of pathogen signatures among our dataset (p -value = 0.09, Fisher Exact test). In line with those results, none of the 14 pig molars (brachydont, although pig canines are hypsodont) produced robust hits.”

Discussion, line 466: “However, it is important to bear in mind that tooth morphology differs between species; for example, humans, dogs, and pigs (except the pig tusks/ canines) have brachydont teeth with enclosed pulp chambers, while sheep, goats, and cattle have hypsodont teeth, which do not have enclosed pulp chambers and often have open roots. While we do not observe a statistically significant difference in the number of robust bacterial hits obtained from these two types of teeth, more research is needed to understand how tooth morphology affects the preservation and recovery of authentic ancient pathogen DNA.”

18. 2c) The identification of *S. enterica* within the studied assemblage is briefly explored as a case study in the Discussion section, but the results are not presented or addressed in the Results section, apart from lines 283-285: “The pathological changes in the mandibles that produced bacterial hits included periodontitis, abscesses, and antemortem tooth loss, with genomic signatures for *Bordetella*, *Corynebacterium*, *Enterococcus*, *Salmonella*, *Staphylococcus*, and *Streptococcus*.”. The presence of *S. enterica* should be more explicitly reported and contextualized within the Results

section, especially given the importance of this pathogen today.

We have restructured the *Salmonella* results to include the sample numbers, sites and, where applicable, the pathologies and expanded the discussion.

Line 263: “Bacterial hits from four samples passed our stringent authentication criteria: AZP-056, a sheep tooth from Petreni (Romania), AZP-116, a sheep mandible with periodontal disease from Pietrele, AZP-148, a dog tooth from Pietrele, and AZP-277, a sheep femur with arthropathies from Tilla Bulak.”

Discussion, line 473: “We recovered several bacterial hits from our dataset including the detection of *S. enterica* signatures, a group of bacteria responsible for gastroenteritis or systemic disease in humans and animals in four specimens from three sites across Eurasia covering a period between 4,000 to 8,000 years ago. Interestingly, two of the four signatures are from teeth, one is from a pathological mandible, and one is from a lesion on a femur, pointing at additional skeletal material that may serve as a source for ancient *S. enterica* DNA. The emergence of human salmonellosis has previously been linked to the Neolithisation of Eurasia and likely has a zoonotic origin. Although the screening dataset did not provide sufficient genome coverage for further analysis of this bacterium, future deep sequencing or target enrichment capture may provide more clarity on the validity and relation between these animal-derived hits and known ancient and modern *S. enterica* diversity.”

19. 3) I understand that the study targets species pathogenic to humans and animals, including opportunist pathogens able to colonise hosts without causing disease. As stated in the Materials and Methods, lines 546-550: "A list of taxa of candidate pathogen species and genera can be found in (Table S3), which includes known zoonotic pathogens and genera, along with additional animal-specific pathogens. We removed the genus node of *Bacillus*, *Clostridium* and *Brucella* due to closely related non-pathogenic species highly abundant in soil that complicates the identification of positive samples." However, I feel further clarification is needed regarding the selection and curation of the reference database used for bacterial, viral, and parasitic taxa.

Thanks for pointing out the need for more information in this section. Our approach was to adapt the widely used screening list of HOPS to our current investigation of zoonotic remains. We performed a literature search to extend this list to include as many animal and human pathogenic species as possible, including for example the subsequently identified pathogen *Erysipelothrix rhusiopathiae*. We note that this list is not exhaustive. We have added explicitly the information on the development of our screening list to the methods section.

Line 616: “A list of taxa of candidate pathogen species and genera can be found in (Table S3); this includes the HOPS screening list of known zoonotic pathogens and genera, and was extended following a literature search with additional zoonotic and animal-specific pathogens. Note that the species list is not exhaustive. We removed the genus node of *Bacillus*, *Clostridium* and *Brucella* due to closely related non-pathogenic species highly abundant in soil that may result in false positive signatures. Following this, we checked all entries for overlap with the NCBI taxonomy database and removed non-findable names. We removed any duplicate entries from the HOPS list.”

20. As the authors used the HOPS pipeline, was the HOPS default pathogen list (https://github.com/rhuebler/HOPS/blob/external/Resources/default_list.txt), which comprised 356 entries, also initially used and further treated and/or complemented with other species of interest? Table S3 lists 266 entries. Note that the HOPS default pathogen list includes some duplicate entries, taxa classified only at the family or genus level without a corresponding species-level reference genome, as well as certain sub-strains and incomplete genomes. Were such entries filtered out or otherwise treated differently during the analysis?

As mentioned above, we did base the screening list on an extended HOPS default list. We compared all entries to the NCBI taxonomy to ensure that they were findable in the database, removing any which were redundant or not findable due to lack of certain species names not in the NCBI taxonomy.

21. Moreover, while Table S3 lists the species of interest to the study, it does not specify the exact reference genomes used or their accession numbers. This information needs to be added for reproducibility purposes.

The HOPS pipeline considers all hits to reference genomes included in the MALT database which are on the taxonomic nodes defined in Table S3. The included genomes are available through the project's github website, which is also referenced in the Screening pipeline methods section. Together these data can be used to find the exact reference genomes for each species of interest. Please see the github repo: pathogen_screening/database_info/all_metadata_database_genomes.tsv (size 17Mb) for the list.

22. 4) In the section on ‘Interpretation of DNA evidence in its palaeopathological context’, the authors point out that 23.3% of all bones with lesions produced robust bacterial hits. They then mention that the “fact that no ancient pathogenic bacteria were authenticated from the 25 bones sampled without any detectable lesions (p-value<0.005, Chi square test) emphasizes the advantage of palaeopathological investigations for prioritisation of specimens.”

Firstly, I think the difference in sample size (189 bones with visible palaeopathological lesions versus 25 bones with no visible palaeopathological lesions) needs to be clearly acknowledged. If the primary aim of the paper was to test and evaluate the advantage of using skeletal elements with pathological lesions compared to samples without, a more balanced dataset would have been needed. The authors recognise this lines 424-426: “Future investigations explicitly designed with comparable amounts of pathological and non-pathological specimens per site will be instrumental to further corroborate our findings.” But it needs to be explicitly stated beforehand.

We have toned down the sentence and also like to point out our changes to emphasize more explicitly our sampling strategy of bones with pathological lesions or teeth to increase the probability to identify pathogen signatures (comment 14).

Line 295: “Although this study is biased towards bones with pathologies, the fact that no ancient pathogenic bacteria were authenticated from the 27 bone samples without any macroscopically detectable

lesions (p-value<0.00035, Fisher Exact test) emphasises the advantage of palaeopathological investigations for prioritisation of specimens.”

23. Secondly, while the reported Chi-square test yields a significant result ($p < 0.005$) for the samples without any detectable lesions, the small sample size and the fact that no ancient pathogenic bacteria was authenticated ($n=0$) suggest that a Fisher’s exact test may be more appropriate in this context. I would recommend reporting results from a Fisher’s test as a robustness check, or at least discussing the statistical limitations of using Chi-square under these conditions, especially if this is part of the main findings of the paper, which it appears to be based on the second paragraph of the discussion (lines 397-399; lines 412-414).

We agree that given the low sample size for several comparisons in the contingency table the Fisher Exact test is more appropriate. We changed all statistical analyses accordingly. This adjustment of tests did not change any of the statistical significance of the results or interpretation presented.

24. 5) Lines 486-487: “ Their different chronologies made it possible to capture changes over time and, in particular, the timing of pathogen transmission in the human–animal relationship”. Similarly to point 4, this is misleading considering the majority of the samples come from the Bronze Age (65%) and no statistical tests were undertaken between the different time periods to support this statement. This also leads to confusion in relation to the aim of the paper, which according to this statement, was to evaluate if increases or decreases of pathogens could be observed over time (this is also further complicated by the fact that there are no reports on all the pathogens identified for each site, if this was indeed the aim of the paper).

We agree that this statement is confusing as indeed our data is insufficient to capture epidemiological processes, like pathogen abundance over time. Instead by probing specimens from different time periods, we increase the chance to cover periods for the emergence of individual pathogens. That is important, because despite preliminary evidence pointing at the Bronze Age period for an overall elevation of zoonotic disease, other periods may also be linked with individual pathogens. Accordingly, we have re-worded that sentence.

Line 526: “Second, their different chronologies may capture pathogen emergence within animal populations over time.”

25. 6) The term “One Health” is listed as the first keyword, and is briefly explained in the opening paragraph of the introduction. However, this research does not engage with the One Health framework. While it includes both domestic and wild animals and targets zoonotic pathogens, the analysis and discussion are centred exclusively on domesticated species, with no reference to human or environmental health. As such, I recommend removing “One Health” as a keyword as it is misleading, unless the authors expand the discussion to more clearly align the study with this interdisciplinary framework.

We agree with the reviewer that the One Health aspect of our work benefits from a more prominent integration into the manuscript. The One Health concept recognizes the interconnection between human, animal, and environmental health and recently began to receive attention from the ancient DNA

community. Expanding ancient pathogen DNA research into the zooarcheological record promises to contribute to the One Health concept, through the identification of zoonotic and reverse zoonotic pathogens - microbes shared by humans and animals - and understand their temporal and geographic spread, reservoirs and the genetic changes that led to spill over into human populations. Our study taps into this emerging research field and provides a zooarcheological dataset for integration into the One Health paradigm that targets zoonotic pathogens. As such we think the “One Health” keyword is justified but moved it to the last position. We have accordingly aligned the contribution of our study more explicitly with the One Health concept.
See also comment #4.

Introduction line 96: “While screening for microbes in large genomic datasets generated for ancient human population studies has become reasonably routine, the same cannot be said for ancient faunal datasets. As a result, limited information is available concerning disease reservoirs in ancient and historic animal populations - posing a barrier to integrate the One Health concept and ancient DNA.”

Discussion Line 513: “Hence, this study is an important step towards exploring ancient pathogens in the zooarchaeological record, necessary to expand the One Health concept into the past, which holds the promise to elucidate the reservoir and host range of zoonotic pathogens, the geographic and temporal spread, and the genetic mechanisms enabling the evolutionary adaptation towards the human host.”

26. 7) The first two lines of the Materials and Methods mention: “The study materials were animal bones and teeth. An initial archaeozoological assessment had been carried out for all the faunal assemblages.” This is very limited information for a Materials and Methods section. The study material needs to be described in greater detail here, including the number of bones and teeth analysed, and the time periods covered - similar to what is outlined in the Results section.

We have revised the Materials and Methods section to include the number of bones and teeth, time periods, and geographical ranges of the study material. Moreover we updated Supplemental Table 2, now including a description of the location and type of paleopathology identified and tested here.

27. Furthermore, as the zooarchaeological analysis represents the baseline of this study - providing species identification through morphology and recording bone surface modifications (e.g. taphonomic marks, cut marks, and of course pathological lesions), a more comprehensive description of the zooarchaeological methodology is essential. I acknowledge that there are references to some manuals, but given how key the zooarchaeological analysis is and for purposes of clarity and reproducibility, these should be clearly detailed.

A more comprehensive description of the zooarchaeological methodology has been added in an improved version of the manuscript.

Line 532: “ The analysis was based on palaeopathological literature and comparative collections stored in the Institute of Geology at Adam Mickiewicz University, Poznań and German archaeological Institute in Berlin. Cases of pathology were classified according to Bartosiewicz and Gal (2013). The recording

system for cases of pathology followed the protocol of Vann and Thomas (2006), and included recording all cases of lesions, without focusing on extreme or rare ones. All locations of the palaeopathological lesions presented in Table S2 were recorded according to the *Nomina Anatomica Veterinaria*.”

Minor comments :

28. 8) Lines 214-216, the authors state: “From the 346 skeletal elements, we produced a total of 357 DNA extracts. These included 20 subsamples collected from 11 bones as well as nine teeth collected from four individuals”.

Having looked through TableS2, I have identified 11 teeth from four individuals:

KOK-009 (3 teeth)

AUTAW111-004 (4 teeth)

AUTAW85-003 (2 teeth)

AUTAW85-023 (2 teeth)

Thank you for noticing this. The correct numbers are 20 subsamples from nine bones and 11 teeth. We have updated the text and the Notes column in Table S2 to clearly state which samples originate from the same individual or are subsamples of the same bone.

29. 9) Lines 216-217: “Samples were sequenced on the Illumina platform, which, apart from five failed samples, generated between 2,324,302 and 90,823,912 sequencing reads per sample (Table S2).”

Only 4 samples have N/A, unless sample ID 353/17.F1 is also a fail, in which case I would clarify it in the notes column.

We do indeed consider sample 353/17.F1 as a failed sample as it only produced 846 sequencing reads. We have updated the Notes column in Table S2 to clarify which samples failed.

30. 10) Line 304: “ No other site was enriched for samples with pathogen signatures.”

The term 'enriched' within a metagenomic and palaeogenomic study is misleading. I would use another term.

We improved the sentence clarity.

Line 332: “No other sites had disproportionate pathogen recovery that rose to statistical significance.”

31. 11) Lines 338-341: “Ascertaining 11,970 mutations among the outgroup *E. tonsillarum*, a previously human-derived medieval *E. rhusiopathiae* genome and 42 *E. rhusiopathiae* genomes isolated from a diverse set of contemporary hosts, [...]”

There are however 3 previously human-derived medieval *E. rhusiopathiae* genomes on the phylogenetic tree (Figure 5), and a total of 7 in Table S4.

Furthermore, in the caption of Figure 5: ”Among 42 *E. rhusiopathiae* genomes and a single *E. tonsillarum* (outgroup) genome we ascertained 11,970 mutations. For four ancient genomes (AZP-012 and three previously published human-derived genomes) the nucleotide call was inferred for each of the mutations.”

It is here confusing which dataset was used for the ascertainment process. And if performed on only a single human-derived sample, as mentioned in the text, could the author provide the corresponding sample ID.

We thank the reviewer for pointing out the ambiguity in this section and the inconsistency across the main text and the figure caption. In short, the analysis is split into two parts. First, we ascertain phylogenetically informative SNPs within the species/genus using an ad hoc set of public datasets (modern genomes as well as published ancient *E. rhusiopathiae* genomes). Notably, in Table S4 we report all genomes initially included, despite some of them eventually dropping out due to our filter regime. Second, within all ancient genomes (our study alongside published ancient genomes) we estimate the genotype for the ascertained position in order to place them in the phylogeny excluding those not meeting our filter criteria. For complete transparency and reproducibility, we report all modern and ancient genomes initially used, despite some of them not being included in the final phylogeny due to our stringent quality control. We have now improved the text in the results and caption and completely rewritten the method section to explain explicitly what has been done and which genomes have been filtered at each step. Moreover, we included information (incl. sample ID) about which genome contributed to which analysis to Table S4.

Results line 369: “We identified 11,970 phylogenetically informative SNPs among 42 *E. rhusiopathiae* genomes isolated from a diverse set of contemporary hosts, including domesticates and marine mammals, a published human-derived medieval *E. rhusiopathiae* genome with sufficient coverage (ldo050) 39 and the outgroup *E. tonsillarum* (Table S4).”

Results line 391: “In order to place the ancient genomic information into the modern genetic diversity, we leveraged 20 public genomes of *S. lutetiensis*, *S. infantarius*, *S. equinus*, *S. salivarius* and *S. gallolyticus* isolated from different sources across Eurasia. All modern genomes except the *S. salivarius* representative passed QC after alignment to the *S. lutetiensis* reference genome (Methods) (Table S4). We identified 92,526 phylogenetically informative positions across the modern representatives.”

Methods section page 687: “For *E. rhusiopathiae*, we utilised all RefSeq genome assemblies available in August 2024 and additional representatives from *E. tonsillarum*, *E. inopinata* and *B. extructa* as outgroups, in line with a previous investigation of the ancient and modern diversity of the pathogen 39 (N=50, including reference genome). For *S. lutetiensis*, we utilised an ad hoc selection of genomes with raw sequencing data available from ENA based on the species and related species in the *S. bovis* subgroup including *S. infantarius*, *S. salivarius*, *S. gallolyticus*, and *S. equinus*, aiming to include a representative sampling of the geographic and host diversity in the species, despite inconsistencies in the reporting of this information across various sequencing records (N=21, including reference genome) (Table S4).

...

Additionally, for *E. rhusiopathiae*, we reanalyzed eight additional publicly available ancient metagenomes previously identified as positive for the species in Iberia from the past 1000 years 39.

...

We utilised the following quality control thresholds: First, we only considered mapped genomes with a global coverage above 3X for the initial identification of phylogenetically informative mutations. For *E. rhusiopathiae*, this removed 14 genomes, four *E. rhusiopathiae* ingroup genomes which are suppressed in

the NCBI database due to failed taxonomy match checks as of 10/2025 (Table S4), and the outgroup genomes of *E. inopinata* and *B. extracta*, and eight ancient genomes which fell below 3x coverage. For *S. lutetiensis*, this removed the outgroup *S. salivarius* genome.”

32. 12) Lines 360-361: “In order to place the ancient genomic information into the modern genetic diversity, we leveraged genomes of *S. lutetiensis*, *S. infantarius*, and *S. gallolyticus* isolated from different sources across Eurasia (Table S4).”

To enhance clarity, could the authors state the total number of genomes used for the ascertainment process, even if this information is already included in the figure caption and Table S1?

Table S4 lists 21 genomes in addition to the three ancient samples from this study. These include *S. lutetiensis*, *S. gallolyticus*, and *S. infantarius*, but also include the outgroup *S. salivarius* and five *S. equinus* genomes. This differs from what is mentioned in the text. Could the author clarify this?

We thank the reviewer for pointing out this inconsistency and lack of information, which we addressed for the *S. lutetiensis* analysis alongside the *E. rhusiopathiae* analysis. Please see our reply to comment 31 above.

33. 13) Lines 394-396: “In this study, we applied a targeted approach to explore the suitability of zooarchaeological remains that primarily displayed palaeopathological lesions, which could have resulted from infection, for genomic palaeomicrobiological analysis”

In the context of metagenomic studies, the term ‘targeted approach’ can be misinterpreted as referring to the molecular targeting strategy (e.g. enrichment and capture), rather than the intended meaning of targeting particular skeletal elements or specimens. I recommend rephrasing this sentence to avoid ambiguity.

Thank you for pointing out the possibility of misinterpretation in this section. For clarity, we have adjusted the wording to “selective sampling approach” both here and elsewhere in the manuscript

Line 429: “In this study, we combined palaeopathological analysis and ancient genomics to explore whether a selective sampling approach focused on palaeopathological lesions indicative of infection or teeth provide a useful starting point for ancient DNA sampling for faunal palaeomicrobiological analysis.”

Typos/Wording :

34. Line 58: Should it not be ‘Neolithic’ rather than ‘prehistory’?

35. Line 138: “The sites cover periods dating from the Neolithic to the Medieval period [...]”

36. Line 140/141: “[...]because human-derived ancient zoonotic pathogen genomes have been repeatedly identified in the literature in specimens from the Bronze Age.”

38. Line 305: “While we observe...[...].” remove comma

39. Line 316: “In summary, despite...”

40. Line 427: “Bone remodelling is generally a slow process 50, although some acute infections [...]”
remove semi-colon, add comma

41. Line 436-437: Dot is at the beginning of line 437, needs to be at end of 436.

All typos and suggested changes to the wording have been fixed.

Reviewer #3 (Remarks to the Author):

This is an excellent piece of work by Felix Key's team. A large part of this paper's novelty lies in showcasing querying zooarchaeological assemblages for pathogen DNA, and also this paper offers a fresh perspective on focusing on pathogens that are not exclusively human, nor centered around historic human epidemics. They have employed a robust set of established methods, as well as developed custom pipelines, to metagenomically identify pathogens in ancient DNA data and perform robust phylogenetic placements of low coverage samples in modern phylogenetic trees. I highly recommend this paper gets published.

Here is a list of minor comments that could help polish this manuscript even more:

44. Intro

Lines 71-79: Even though I understand what the authors are trying to communicate this piece of text feels a bit choppy and "rushed" in terms of introducing all these ideas. Maybe moving the sentence starting with "For instance, the 1918 flu pandemic ..." right before the previous sentence ("Such zoonotic spillover events can ...") would improve the fluidity of that section.

Thank you for this comment. We have revised the introduction to improve clarity and readability in response to this comment as well as those from the other reviewers.

Line 77: “Infectious diseases are a major health concern responsible for an estimated 13.7 million deaths worldwide in 2019. Approximately 60% of human pathogens are believed to have originated from animals with dramatic consequences throughout history. For instance, the 1918 flu pandemic likely had an avian origin, while outbreaks of plague, which are transmitted from rodent hosts via flea vectors, led to the Black Death in 1346–1353 CE. Although Palaeolithic hunter-gatherers came into contact with diseased animals, many zoonotic diseases were likely introduced into human populations following the introduction of farming during the Neolithic Transition starting around 12,000 years ago.”

45. Results

Lines 150-151: The authors mention in that figure legend that "Note that the Bronze Age covers different absolute chronological horizons within different geographical locations." This is a really important point and it would be worth including in the main text, as this might not be an apparent piece of info for readers not accustomed to ancient DNA and/or scientific archaeology.

This is a good point. We have now stated in the main text that the time periods align with material culture development rather than absolute chronology.

Line 159: “These periods took place at different absolute chronological horizons due to spatiotemporal differences in material culture development within discrete geographical locations.”

46. Lines 279-281: This is a cool and reasonable result. Could the authors maybe provide an explanation/speculation for the reason in the discussion ? e.g. could it be because these bones are denser than others, or could it be because these are the more frequently preserved skeletal elements when buried ?

Indeed multiple explanations may underlie this observation. We agree with the reviewer that the presence of pathogens within the zooarcheological record will depend on bone morphology/density as well as that certain bones are more abundant within the animal skeleton, and thus the archeological record. Moreover, other factors at play likely include variation in bone vascularization necessary for pathogen deposition and whether or not diseased animals entered the archeological record similar to non-diseased individuals. Our specimen collection is biased towards bone types available, i.e. abundant specimens collected during excavations, and the identification of pathologies, which subsequently biases the identification of pathogen signatures. In conclusion, our study is not powered to provide sufficient evidence for either of the explanations, as such we refrain from speculation within the manuscript.

47. Lines 294-296: This is a rather interesting but unexpected results. It is true that although *E. coli* and *Enterococcus* are mainly gut commensals, they can be found in other body parts or even free-living in the environment. Finding them in ribs would imply (at least naively) either acute bacteremia or a lung infection (which is rather unusual). Could the authors elaborate a tad more on that result and its interpretation, since they explained very well in their discussion the difficulty of distinguishing a true ancient pathogen from "ancient-like" contaminant species that might have colonised the bone after the host's death ?

Thanks for the suggestion, we agree that the finding of both gut commensals is interesting. However, as the reviewer mentions, bacteria like *E.coli* and *Enterococcus* are highly successful within the environment, oxygen tolerant, and prominent within the microbiome of humans and animals, rendering them as species with a high potential for contamination. For this reason, and in the absence of phylogenetic authentication despite the authentic signature, we prefer to avoid speculation about any particular disease the bacteria may or not have caused among the sampled animals. We acknowledge this aspect in the results and the discussion.

Line 323: ”Other species signatures recovered from palaeopathological lesions on ribs included species of *Enterococcus* and *E. coli*, both potentially contaminants due to their abundance in human microbiomes.”

Line 501: “While such analyses were not possible for all potential pathogens detected here, they are important in order to distinguish between signatures of authentic endogenous pathogens. Especially opportunistic pathogens present in the human microbiome pose a risk, as they may have been deposited

during excavation and subsequent handling, and could have developed degradation patterns reminiscent of authentic ancient DNA while in storage.”

48. Lines 313-315: This is a really interesting result. Could the authors elaborate on it in the discussion and provide a hypothesis of why that happened ?

Our sampling strategy is designed to cover many different sites to avoid biases that may be present at a given site. As a result, we are not able to make a qualitative statement about why the Tilla Bulak site is overrepresented among pathogen-positive hits. We explicitly acknowledge this caveat in the discussion.

Line 445: “Moreover, we observe variation in DNA preservation across sites presumably due to differences in depositional practices and environmental conditions, but potentially also influenced by specimen handling and storage, all of which may contribute to the differences observed in recovered pathogen DNA. In addition, different methods for DNA laboratory processing as well as sequencing depth might influence pathogen DNA recovery; in this study, the only site, Tilla Bulak, where the number of robust pathogen hits rose to statistically significant levels was processed using the single-stranded Santa Cruz protocol for library preparation.”

49. Figures 5 and 6: I am curious why the authors chose to include modern genomes that did not have full metadata on the host and the country of origin ? Was it because there were not enough high quality modern genomes with their metadata filled in that fit their criteria ? If so it could be really helpful to make a note of it.

In general, we sought to explore a range of geographic localities and host sources, despite some genomes not having complete metadata for the understudied species group.

For *E. rhusiopathiae*, we analyzed the available RefSeq genomes as of August 2024 for the species to investigate the phylogenetic placement of our ancient genomes relative to their isolation, in line with a previous publication (PMID: 39196943).

For genomes of *S. lutetiensis* and related species, the availability of metadata is overall poor. We note that two of the genomes without location metadata include the reference genome of the species (D02) in addition to the type strain of *S. lutetiensis*.

We have expanded the methods section on our selection of genomes for the phylogenetic comparisons for clarity.

Line 687: “For *E. rhusiopathiae*, we utilised all RefSeq genome assemblies available in August 2024 and additional representatives from *E. tonsillarum*, *E. inopinata* and *B. extructa* as outgroups, in line with a previous investigation of the ancient and modern diversity of the pathogen (N=50, including reference genome). For *S. lutetiensis*, we utilised an *ad hoc* selection of genomes with raw sequencing data available from ENA based on the species and related species in the *S. bovis* subgroup including *S. infantarius*, *S. salivarius*, *S. gallolyticus*, and *S. equinus*, aiming to include a representative sampling of

the geographic and host diversity in the species, despite inconsistencies in the reporting of this information across various sequencing records (N=21, including reference genome) (**Table S4**). For both *E. rhusiopathiae* and *S. lutetiensis*, the listed genomes were included to optimise the genetic representation due to the overall limited number of available genomes and despite the inconsistencies in available metadata.”

50. Methods:

The "Pathogen phylogenetic reconstruction", although detailed could be written in a simpler/clearer way so that it is easier to follow through the individual steps the authors took.

We thank the reviewer for pointing out the lack of clarity in this section, which was also raised by reviewer 2 (comment 31/32). We have rewritten the entire methods paragraph on “Pathogen phylogenetic reconstruction”). Please see our reply to comment 31 or the revised methods section in the manuscript.

Reviewer #3 (Remarks on code availability):

51. The GitHub repo linked to this manuscript contains all the necessary code and scripts to reproduce the analyses presented in this paper

Yes.

Response to reviewers

REVIEWERS' COMMENTS

Reviewer #1 (Remarks to the Author):

The authors have responded appropriately and precisely to my previous comments and I am happy to support the manuscript being published. It is an excellent piece of work, which will be of interest to a broad range of researchers.

We highly appreciate the positive review of our revised manuscript.

Reviewer #2 (Remarks to the Author):

The revisions made to the manuscript have greatly improved both its clarity and coherence, with the overall aim of the study now explicitly stated. By combining palaeopathological and genomic analyses across a comprehensive dataset of 346 skeletal elements from both domesticated and wild animals, recovered from 34 Eurasian sites spanning the last 7,000 years and a wide range of environments, the authors provide well-supported recommendations for sampling strategies for ancient pathogen DNA research within the zooarchaeological record. Archaeological faunal specimens remain a resource vastly understudied within the field of palaeomicrobiology, and their overall potential has remained unaddressed until now. As such, this study represents an essential guide for future palaeopathological and epidemiological investigations based on ancient faunal remains.

The authors identified several bacteria taxa, including *S. enterica*, *C. burnetii*, and *P. gingivalis*, and present two case studies focusing on *E. rhusiopathiae* and *S. lutetiensis*, in which both the ancient nature of the genomic data is authenticated, and the evolutionary relationships between these selected low-coverage ancient genomes and their present-day relatives are examined. Overall, all major findings are addressed in sufficient detail, and limitations are acknowledged where appropriate.

The results and discussion are well aligned with the study's original aim. The discrepancies between the text and the tables have been addressed, and the manuscript reads with a clear and cohesive narrative. I thank the authors for such careful and thorough revision. I would also like to commend the high level of transparency, particularly regarding data and code availability, which is essential for ensuring the reproducibility and reliability of the work.

Overall, all comments have been satisfactorily addressed, and I believe the manuscript represents a strong and valuable contribution to the field and is suitable for publication. I have no major comments, only three minor points intended more for clarification.

1. In the abstract, it is mentioned that “68 signatures of ancient (opportunistic) pathogens” have been identified (line 69). Could the authors clarify what this refers to? According to Figure 4, I understand that 29 bacterial pathogens have been identified across the 55 libraries that produced high-confidence bacterial hits, of which 24 libraries displayed hits from multiple bacteria.

Thanks for identifying this mistake. Indeed we identified a total of 116 signatures from 29 bacterial pathogens across 55 libraries/samples. We carefully combed through the manuscript to make sure all reported numbers are correct.

2. Lines 260-261, lines 305-308 and lines 676-677, the authors refer to Figure 3 when discussing the identification of signatures from various bacteria including *E. rhusiopathiae*, *S. lutetiensis*, *Bordetella*, *Corynebacterium*, *Enterococcus*, *Salmonella*, *Staphylococcus*, and *Streptococcus*. Yet Figure 3 illustrates examples of pathological lesions sampled in this study and which produced pathogen signatures for the screening, but it does not explicitly indicate the bacterial taxa. I was wondering if the authors intended to refer to Figure 4, which clearly presents the full range of identified bacterial taxa?

Thanks for pointing out this referencing mistake. We have fixed that now. All referenced tables and figures are now correct.

3. In Table S4, seven ancient *E. rhusiopathiae* Iberian genomes are listed. However, in the main text (line 699), it is mentioned that eight additional publicly available ancient metagenomes were reanalysed. I think there may be a discrepancy?

Indeed only seven ancient *E. rhusiopathiae* Iberian genomes from Rodríguez-Varela et al. (2024) were available and used. We corrected the text accordingly.